# Estimating velocity distribution and flood discharge at river bridges using the entropy theory. Insights from Computational Fluid Dynamics flow fields

Farhad Bahmanpouri[1†], Tommaso Lazzarin[2†], Silvia Barbetta[1], Tommaso Moramarco[1], Daniele P. Viero[2]

[1]Research Institute for Geo-Hydrological Protection, National Research Council (CNR), Perugia, 06128, Italy
[2]Department of Civil, Environmental and Architectural Engineering, University of Padova, 35131, Italy

[†] F.B. and T.L. equally contributed to the study.

*Correspondence to*: Daniele P. Viero (daniele.viero@unipd.it)

**Abstract.** Estimating the flow velocity and discharge in rivers is of particular interest for monitoring, modelling, and research purposes. Instruments for measuring water level and surface velocity are generally mounted on bridge decks, and this poses a challenge because the bridge structure (e.g., piers and abutments) can lead to perturbated flow fields. The current research aims to investigate the applicability of the entropy theory to estimate the velocity distribution and the discharge in the vicinity of river bridges. To this purpose, a Computational Fluid Dynamics (CFD) model is used to obtain three-dimensional flow fields along a stretch of the Paglia River (central Italy), where a historical multi-arch bridge strongly affects flood flows. The input data for the entropy model include the cross-sectional bathymetry and the surface velocity provided by the numerical simulations. A total of 12 samples, including three different flow conditions for four cross-sections, one upstream and three downstream of the bridge, are considered. It is found that the entropy model can be reliably applied upstream of the bridge, also when forced with a single (i.e., the maximum) value of the surface velocity, with errors on total discharge below 13% in the considered case. On the contrary, downstream of the bridge, the wakes generated by the bridge piers strongly affect the velocity distribution, both in the spanwise and in the vertical directions, and for very long distances. Here, notwithstanding the complex and multimodal spanwise distribution of flow velocity, the entropy model estimates the discharge with error lower than 8% if forced with the river-wide distribution of the surface velocity. The present study has important implications for the optimal positioning of sensors and suggests the potential of using CFD modelling and entropy theory jointly to foster the knowledge of river systems.

## 1 Introduction

Velocity and discharge measurements in rivers are fundamental for monitoring, modelling, and research purposes (Depetris, 2021; Di Baldassarre and Montanari, 2009; Dottori et al., 2013; Gore and Banning, 2017; Herschy, 2009). Unfortunately, measuring river discharge can be very challenging due to different reasons, for example in the case of intermittent rivers typical of semi-arid regions, of flash floods in mountain areas, of flood flows involving wide floodplains, of freshwater flows affected

by saline tidal intrusions in estuaries, etc. While monitoring river discharge on the ground has definite advantages (Fekete et al., 2012), the use of traditional methods such as current meters and ADCPs is generally expensive, time-consuming, and risky for operators, mainly during severe flow conditions, and such methods are not applicable in remote and inaccessible locations. Different techniques can be used to measure the surface velocity, also during severe flood conditions, including Large-Scale Particle Image Velocimetry (LSPIV) (Eltner et al., 2020; Jodeau et al., 2008; Le Coz et al., 2010; Muste et al., 2011, 2014), Space-Time Image Velocimetry (STIV) (Fujita et al., 2007, 2019), Infrared Quantitative Image Velocimetry (Schweitzer and Cowen, 2021), and other methods based on the use of either terrestrial or Unmanned Aerial System sensors (Bandini et al., 2020, 2021; Herschy, 2009).

Indirect methods have then been proposed to estimate the flow discharge using this kind of remote sensed data (Bogning et al., 2018; Fekete and Vörösmarty, 2002; Spada et al., 2017; Vandaele et al., 2023; Zhang et al., 2019). The flowrate is generally obtained by applying suitable velocity coefficients to estimate the depth-averaged velocity or by integration of a hypothetical flow velocity distribution in the cross-sectional area. The key point is thus estimating the depth-averaged velocity, or its full cross-sectional distribution, starting from surface velocity data, a process whose reliability depends on the (un)evenness of the actual velocity distribution.

In natural rivers with large cross-sections, the streamwise velocity typically shows a logarithmic vertical distribution, mainly determined by the bottom roughness. According to field data, the maximum velocity is found just below the free surface and gradually decreases towards the bed (Franca et al., 2008; Guo, 2014). However, plenty of factors contributes to making the velocity distribution irregular. For instance, channel bends and deformed bathymetry produce large-scale secondary currents (Constantinescu et al., 2011; Lazzarin and Viero, 2023; Yang et al., 2012), and the presence of banks and of discontinuities of bed elevation in the spanwise directions can generate secondary currents of the second kind because of turbulence heterogeneity (Nikora and Roy, 2011; Proust and Nikora, 2020), which all increase the three-dimensionality of the flow field and alter the vertical and spanwise distribution of the flow velocity.

The presence of in-stream structures, such as bridges characterized by the presence of piers and/or of lateral abutments, can induce further alterations on the flow field (Laursen, 1960, 1963), producing complex and rapidly varying flow patterns, with the formation of strong three dimensional flow structures (Ataie-Ashtiani and Aslani-Kordkandi, 2012; Chang et al., 2013; Salaheldin et al., 2004). Secondary currents in the cross-section transport low momentum fluid from lateral region to the center of the channel, and high-momentum fluids from the free surface toward the bed (Bonakdari et al., 2008; Nezu and Nakagawa, 1993; Yang et al., 2004). Systems of vortices with horizontal (horseshoe vortex) or vertical axes (wake vortex) modify the velocity distribution (Kirkil and Constantinescu, 2015; Sumer et al., 1997). The wakes generated by in-stream obstacles and contractions can produce uneven spatial distributions of the water surface elevations close to the bridge, and can propagate downstream of bridges thus altering the cross-sectional velocity distribution for quite long distances (Briaud et al., 2009; Yang et al., 2021). Furthermore, because of particular bridge shape (e.g., arch-piers) and of irregular cross-sections (e.g., compound sections), the flow field may show a marked dependence on the water depth and the flowrate.

Even though the above factors complicate inferring the cross-sectional velocity distribution (and thus the flow discharge) based on surface velocity data in the vicinity of in-stream structures, it has to be observed that measuring instruments such as hydrometers, as well as radar sensors or cameras for estimating the surface velocity, are often mounted on bridge decks for convenience reasons. Notwithstanding the recommendation of installing height gage at the upstream side of bridges (Meals and Dressing, 2008), measuring instruments are often located downstream of bridges, where the flow field unevenness is expected to further complicate the discharge estimation (Kästner et al., 2018). Besides the measurement of the flow discharge, the knowledge of flow field nearby bridges has additional practical implications; the flow velocity is the dominant parameter to study the local scour at a bridge pier, which may result in being responsible for the bridge collapse in some extreme conditions (Barbetta et al., 2017; Federico et al., 2003; Lu et al., 2022). The formation of scours at piers and abutments can be attributed to a significant extent to the flow patterns produced at their immediate vicinity, such as the flow contraction and the large-scale turbulent structures (Cheng et al., 2018; Khosronejad et al., 2012).

One of the most promising methods to estimate the cross-sectional velocity distribution from joint measures of water level and surface velocity is based on the entropy concept. Researchers have widely applied this concept to predict velocity distributions, the flow discharge, and other relevant parameters in open channels (Bahmanpouri et al., 2022b; Bonakdari et al., 2015; Chahrour et al., 2021; Chiu, 1989; Chiu et al., 2005; Chiu and Said, 1995; Ebtehaj et al., 2018; Moramarco et al., 2019; Moramarco and Singh, 2010; Singh et al., 2017; Sterling and Knight, 2002; Termini and Moramarco, 2017; Vyas et al., 2021). Recent applications of the entropic velocity distribution include the case of large meandering channels (Termini and Moramarco, 2020), the estimation of the depth-averaged velocity as a function of the aspect ratio (Abdolvandi et al., 2021), the confluence of the large Negro and Solimões rivers (Bahmanpouri et al. 2022a), and the regularizing of the entropy parameter (Ammari et al., 2022). One advantage of the entropy approach is providing the complete cross-sectional distribution of velocity, whereas other indirect methods for estimating flow discharge only compute the depth-averaged value from the surface velocities at subsections using a fixed reduction coefficient (e.g., Le Coz et al., 2010). Previous studies demonstrated the accuracy of the entropy method referring to undisturbed flow conditions, and also to cases like confluences or low curvature bends characterized by large-scale three-dimensional effects and secondary currents.

The present research is meant to investigate the predictive ability of the entropy theory in estimating the velocity distribution, and hence the streamflow discharge, in the case of complex flow fields generated by the presence of bridges. It is of particular relevance because, as already noted, water levels and free-surface velocities are often measured by instruments mounted on bridges, where the flow-structure interaction can significantly disturb the flow field.

Considering that measuring the cross-sectional velocity distribution in the vicinity of bridges is practically unfeasible in flood conditions, in the present study a three-dimensional Computational Fluid Dynamics (3D-CFD) model is used to obtain physics-based and high-resolution descriptions of the real flow field, for a sufficiently long river segment and for different values of the flow discharge. The CFD-computed surface velocity (either a single value or its river-wide distribution) is used as input for the entropy model, thus simulating the availability of suitable remote sense instruments. Then, the cross-sectional velocity distributions provided by the entropic model are benchmarked against those computed by the CFD model, which allows

assessing the reliability of the entropy model. The exercise is repeated for different cross-sections, both upstream and downstream of the bridge, to investigate the pros and cons of different locations where estimating the discharge, thus to provide applicative guidelines. A reach of the Paglia River, in the central Italy, is chosen as a relevant case study; here, a level gauge

and a radar sensor for measuring the surface velocity are mounted on a historical multi-arch bridge, which produces strong flow-structure interactions.

The present analysis allows providing guidelines for the proper application of the entropy theory and the optimal choice and positioning of measuring instruments, aimed at the reliable estimation of flow discharge in the vicinity of river bridges.

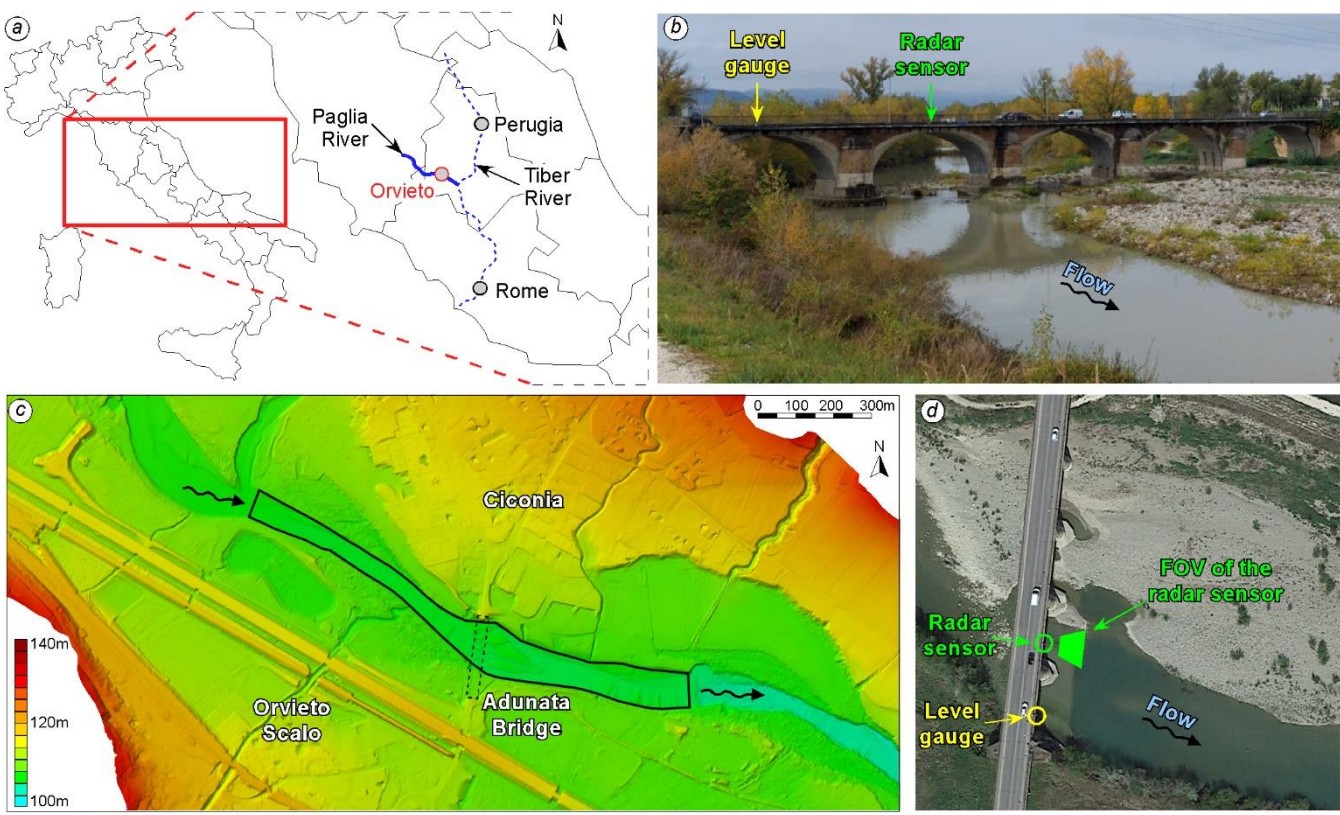

**Figure 1. a) Location of the field site; b) downstream view of the Adunata bridge on the Paglia River during normal flow condition (11.11.2021); c) Digital Terrain Model (DTM) nearby the Adunata bridge (dotted line), with the domain of the 3D CFD model (black line); d) location of the level gauge and of the radar sensor with the field of view (FOV) on aerial image (© Google Earth).**

## 2 Material and Methods

### 2.1 Field Site

The Paglia River, in the central part of Italy (Figure 1a), is a tributary of Tiber River, subjected to severe flooding and high sediment transport. The reach of interest is across the Adunata bridge (Figure 1b), near the town of Orvieto, where the Paglia

River subtends a basin area of about 1'200 km². Here, the average discharge is of 10 m³/s; however, flood discharge can reach estimated values of up to 2'500 m³/s.

The Adunata bridge connects the settlements of Orvieto Scalo and Ciconia, as part of the Italian State Road n.71 (Figure 1c).
It is a masonry multi-arch bridge, with 5 arches ending at four piers on the river bed. On the right-hand side, an abutment sustains the bridge and separates it from the floodplain; on the left-hand side, the bridge deck is supported by the main levee. Close to the bottom, the piers have a roughly elliptical shape with the major axis, aligned with the flow, 15 m long, and the minor axis, orthogonal to the flow, 5.7 m wide. At the bottom, each pier is sustained by an elliptical plinth whose profile is 2.0 m larger than the pier. The center-distance between the piers is 23.2 m. The piers width increases approaching the deck
because of the arches; the deck width is approximatively 10 m.

The main thread of the flow is at the right-hand side of the river, and a large depositional area forms on the left-hand side just downstream of the bridge (Figure 1b). The main channel axis is characterized by a significant curvature, bending to the left at the bridge section (Figure 1c).

## 2.2 Available Data

In the downstream side of the Adunata Bridge, a water level gauge is placed at the center of the first arch, and a radar sensor for measuring the water surface velocity is located at the second arch (Figure 1d). The field of view (FOV) of the radar sensor is also shown in Figure 1d. The time resolution of both the sensors is 10 min.

In addition, a number of flowrate measures and cross-sectional velocity distributions are provided by the Umbria Region Hydrological Service. The flowrate data were collected using a current meter a few tens of meters downstream of the Adunata
Bridge, by wading in the period 2009-2011 (flowrate ranging between 3.3 and 14.3 m³/s), and from the bridge in the period 1995-2010 (flowrate ranging between 16.8 and 147 m³/s); additional flowrate data were collected using an Acoustic-Doppler Current Profiler (ADCP) some hundreds of meters upstream of the bridge in the period 2014-2019 (flowrate ranging between 0.37 and 45 m³/s). The official rating curve for the Adunata Bridge, provided by ARPA Lazio, is based on these measures.

As detailed in the following sections, the rating curve derived from current meter and ADCP data, the water levels, and the
free-surface velocity data collected by the sensors mounted on the Adunata bridge, were used to validate the hydrodynamic numerical models (Sect. 2.3 and Appendix A). The cross-sectional velocity distributions measured with the current meter just downstream of the bridge were used to further assess the spatial variability of the entropy-based velocity distributions, as detailed in Sect. 3.1.

## 2.3 Numerical Model

The commercial CFD software STAR-CCM+ (Siemens) was used for the numerical simulations. It implements the Finite Volume method to compute the flow field on unstructured, Cartesian computational grids. The software has been used and validated in several applications for flow over deformed bathymetry in presence of obstacles (Chang et al., 2013; Kirkil et al., 2009; Lazzarin et al., 2023c) and channel bends (Constantinescu et al., 2011, 2013; Koken et al., 2013). In the present

application, the two-phase Volume of Fluid (VoF) method was used to distinguish water and air in the computational domain
(Hirt and Nichols, 1981). This method was shown to well capture the water surface in complex open channel flows (Horna-Munoz and Constantinescu, 2018; Lazzarin et al., 2023b; Li and Zhang, 2022; Luo et al., 2018; Yoshimura and Fujita, 2020). In the used setup, the model solved the Reynolds-Averaged Navier-Stokes (RANS) equations, in which the stress tensor in the momentum equations is related to the mean flow quantities by adopting the Bousinnesq approximation. The eddy viscosity, $\mu_T$, was determined by solving transport equations for the turbulent kinetic energy, $k$, and dissipation rate, $\varepsilon$, according to the realizable $k$-$\varepsilon$ turbulence model (Shih et al., 1995), which was shown to provide reliable predictions for large-scale complex flows in natural rivers (e.g., Horna-Munoz and Constantinescu, 2018).

The simulations were advanced in time with an implicit, $1^{st}$ order discretization, until reaching steady state conditions. The computational domain reproduced a ~1,100 m long reach of the Paglia River (Figure 1c), centered at the Adunata bridge. The average size of the grid elements was of 1.0 m. Starting 100 m upstream of the bridge and up to 300 m downstream of the bridge, the grid was refined using elements with average length 0.5 m. To well capture the near-wall boundary layer, a prism layer refinement with three layers was used to reduce the wall-normal thickness of the grid cells close to solid boundaries (i.e., the riverbed and bridge structure). The final computational grid was made of ~4 million elements.

A rough-wall, no-slip condition was imposed at the solid boundaries by means of a wall function (roughness height of 0.1 m at the bottom, and of 0.01 m at the bridge surfaces). The upper boundary of the computational grid was treated as a symmetry plane (i.e., slip-condition) for the air-flow. The water elevation at the outlet (i.e., downstream section) was kept fixed in time by imposing a suitable hydrostatic-pressure distribution. The value of the downstream level, for each of the simulated scenarios, was derived from an auxiliary two-dimensional (2D), depth-averaged hydrodynamic model calibrated on available data; the 2DEF model has been used to this purpose (see Appendix A for details on the model and its calibration/verification). A constant-in-time, logarithmic velocity distribution was imposed as upstream boundary condition for the water fraction. For the air fraction (upper part of the numerical domain), zero velocity and zero pressure were imposed at the inlet and at the outlet, respectively.

The 3D-CFD model was validated by comparing the surface velocity computed by the model with those measured by the radar sensor located downstream of the bridge (see the yellow bullets in Fig. A2c,d, in the Appendix A).

## 2.4 Flood events considered in the study

Three different steady flow conditions have been simulated with the 3D-CFD model STAR-CCM+, which correspond to the peak flow conditions of flood events occurred in 2012, 2019, and 2022 as provided by the rating curve for the Adunata Bridge (Table 1). In all the three flow conditions, water flowed in the main channel and over the sediment bars that are dry in the low flow condition of Figure 1b,d. During the most severe flood of 2012, water flowed on the floodplains adjacent to the main river and caused the incipient pressurization of flow below the bridge arches. The preliminary simulation carried out with the 2DEF depth-averaged model showed that, at the peak of the 2012 flood event, 700 m³/s flowed through the floodplain, overflowing the bridge access roads, and 1800 m³/s flowed within the main channel; this last value was used in the 3D-CFD

simulation, which considered only the main channel of the river. The flood events of 2019 and 2022, although being quite ordinary, were the largest floods occurred after the installation of the radar sensor for the surface velocity data (surface velocity data were not available for the 2012 flood).

| Event | Discharge [m³/s] | Return period (years) |
|---|---|---|
| 2012 | 1800 (2500) | 200 |
| 2019 | 450 | 2 |
| 2022 | 160 | 1 |

Table 1. Simulations performed in the present work. The value in brackets indicate the total discharge considering also the flow over floodplains, not considered in the 3D simulations.

## 2.5 Entropy theory

The Entropy theory deals with physical systems that may have a large number of states from a probabilistic point of view. The concept of entropy is used for statistical inference, to determine a probability distribution function when the available information is limited to some average quantities, defined as constraints such as mean and variance. For the application of entropy to streamflow measurements, the pioneer was Chiu (1987), who developed a probabilistic formulation of the cross-sectional velocity distribution in open channels, in which the expected value of the point velocity is determined by applying the maximum entropy principle (Chiu, 1987, 1988, 1989). Using this probabilistic formulation, the velocity distribution is given analytically as a function of the cross-sectional geometry, of the dimensionless entropy parameter, $M$, and possibly of the depth at which the maximum velocity occurs (the so-called dip, $h$). There is a one-to-one correspondence between $M$ and the ratio of mean to maximum velocity in the cross-section, which is defined as the entropic function, $\phi(M)$ (Chiu, 1991). In general, for a given river site, the magnitude of $M$ and, in turn, of $\phi(M)$, mainly depend on hydraulic parameters such as roughness and hydraulic radius, whereas they are poorly affected by the flow discharge (Chiu and Murray, 1992; Moramarco and Singh, 2010). Moreover, $\phi(M)$ is consistently found to be nearly constant at different cross-sections through gauged river sites for different flow conditions (Moramarco and Singh, 2010; Ammari et al., 2022). This is because the value of $\phi(M)$ is associated with geometric and hydraulic characteristics that tend to vary smoothly within a river system (Ammari et al., 2022). The estimation of cross-sectional velocity distribution, $U(x,y)$, developed by Chiu (1989), was later simplified by Moramarco et al. (2004). Using this approach, one can divide the wet cross-sectional area into $N_v$ verticals and determine the entropy-based velocity profile along each vertical as

$$U(x_i, y) = \frac{U_{max}(x_i)}{M} \ln\left[1 + (e^M - 1)\frac{y}{D(x_i) - h(x_i)}\exp\left(1 - \frac{y}{D(x_i) - h(x_i)}\right)\right] \quad i = 1 \ldots N_v \quad (1)$$

where $U$ is the time-averaged velocity, $U_{max}(x_i)$ is the maximum value of $U$ along the $i$th vertical, $x_i$ is the distance of the $i$th sampled vertical from the left bank, $h(x_i)$ is the dip (i.e., the depth of $U_{max}(x_i)$ below the water surface), $D(x_i)$ the flow depth, $y$ is the vertical distance from the bed. The relationship between the entropic parameter, $M$, and the entropic function, $\phi(M)$, is (Chiu, 1989):

$$\phi(M) = \frac{U_m}{U_{MAX}} = \frac{e^M}{e^M - 1} - \frac{1}{M} \quad (2)$$

in which $U_m$ and $U_{MAX}$ are the average and maximum flow velocities within the entire cross-section. It is worth mentioning that $U_m$ represents the expected value of velocity that can be different from the observed mean velocity (Marini and Fontana, 2020). These two values are quite similar in the case of wide rivers (aspect ratio larger than 5). In the present research, considering the large aspect ratio for all cross-sections (Table 2), this hypothesis is valid.

Introducing the variable $\delta(x_i) = D(x_i)/[D(x_i) - h(x_i)]$, the velocity dip, $h(x_i)$, is estimated according to Yang et al. (2004) from the spanwise distribution of $\delta(x_i)$, which is given as

$$\delta(x_i) = 1 + 1.3 \exp\left(-\frac{x_{min}}{D(x_i)}\right) \tag{3}$$

in which $x_{min}$ is the spanwise distance of the $x_i$ vertical from the nearest bank. Note that $h(x_i) = 0$ and $\delta(x_i) = 1$ when the maximum velocity occurs at the free-surface.

In case of gauged cross-sections, $\phi(M)$ can be inferred from measured mean and maximum flow velocities (e.g., with ADCP). For ungauged sites $\phi(M)$ can be estimated as (Moramarco and Singh, 2010):

$$\phi(M) = \frac{\frac{1}{n}R^{1/6}}{\sqrt{g}\frac{1}{k}\ln\left(\frac{D-h}{y_0}\right)} \tag{4}$$

where $y_0$ is the vertical coordinate, taken from the bottom, where the velocity is zero, $k$ is the von Karman constant, $R$ is the hydraulic radius, $n$ is the Manning roughness, $D$ is the maximum water depth, and $h$ is computed with Eq. (3) at the talweg, i.e., where the water depth is maximum. According to van Rijn (1982), $y_0 = 0.065\,\xi\,d_{90}$, where $d_{90}$ is the 90th percentile for grain size and $\xi$ a parameter ranging from 1 to 10 (Ferro, 2003; Moramarco and Singh, 2010).

Whether at a river site only the surface velocities, $U_{surf}(x_i)$, are available, then $U_{max}(x_i)$ can be estimated as (Fulton and Ostrowski, 2008):

$$U_{max}(x_i) = \frac{U_{surf}(x_i)}{\frac{1}{M}\ln[1 + (e^M - 1)\delta(x_i)e^{1-\delta(x_i)}]} \tag{5}$$

For the current research, the methodological steps to estimate the cross-sectional velocity distribution (and hence the flow discharge) using the entropy theory, are as follows. The input data is the river-wide velocity distribution at the free-surface, $U_{surf}$, provided by the 3D-CFD model. When only the maximum value of $U_{surf}$ is used as input, corresponding to the hypothetical case in which only point-sensor data are available, the spanwise distribution of $U_{surf}$ is obtained by applying either a parabolic or an elliptical spanwise distribution (Bahmanpouri et al., 2022a). The velocity dip is computed using Eq. (3). The cross-sectional velocity distribution is then obtained using an iterative procedure, in which $p$ denotes the iteration. At the first iteration, the entropic function, $\phi(M)_{p=1}$, is computed with Eq. (4), and $M_{p=1}$ is computed with Eq. (2). After computing the maximum velocity for each vertical, $U_{max}(x_i)_{p=1}$, with Eq. (5), Eq. (1) allows estimating the entropic velocity distribution in the whole cross-section, $U(x_i, y)_{p=1}$. The following iteration starts by computing the average and the maximum flow velocities, $U_m$ and $U_{MAX}$, from the velocity distribution obtained in the previous iteration, then $\phi(M)_p = U_m/U_{MAX}$, $M_p$ using Eq. (2), $U_{max}(x_i)_p$

with Eq. (5), and the velocity distribution $U(x_i, y)_p$ with Eq. (1). The iterative procedure continues until the error of $\phi(M)_p - \phi(M)_{p-1}$ becomes lower than 0.01. For more details, the reader is referred to Moramarco et al. (2017).

## 3 Results and discussions

The comparison between the entropy-based and the CFD-derived velocity distributions has been performed considering four cross-sections (Figure 2), at a distance of 50 m upstream and 50, 100, and 200 m downstream of the bridge, and the three flood

events of 2012, 2019, and 2022 (see Table 1). The sections just upstream and downstream of the bridge are located at a distance of about $0.45B$ from the bridge, with $B$ the width of the river at the bridge section. This is a short distance, particularly considering that the remote sensors for surface velocity (such as radar, Large Scale PIV, etc.) have their field of view located some tens of meter upstream or downstream of the bridge. The sections far downstream are considered to assess how far the flow field is affected by the presence of the bridge.

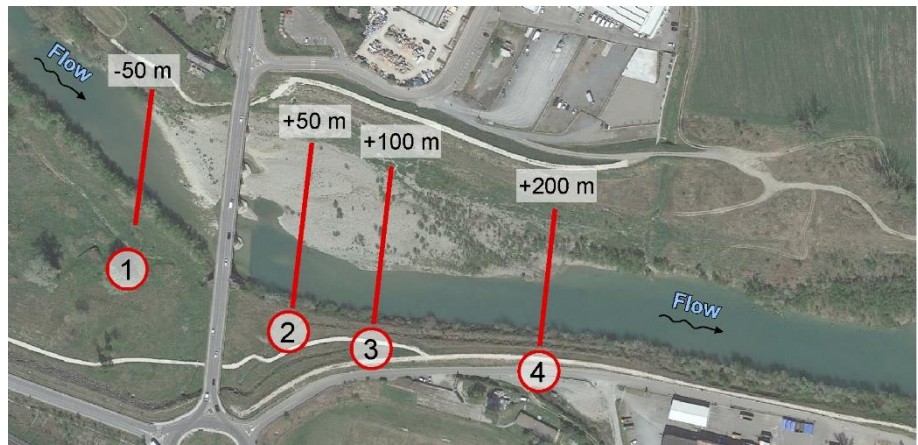

**Figure 2. Location of the Adunata Bridge and of the four selected cross-sections (aerial image from © Google Earth).**

First, the study analyzes the variability of the entropy function, $\phi(M)$, at the four cross-sections, as derived from the cross-sectional velocity distributions provided by both the 3D-CFD model and the current meter measures (Sect. 3.1). Then, in applying the entropy model to estimate the cross-sectional velocity, two different procedures are considered. In the first one,

the entropy model was forced with the river-wide distribution of the surface velocities computed by the 3D-CFD model (this is described in the following Sect. 3.2); in the second one, only the maximum value of the surface velocity computed by the 3D-CFD model was considered as input for the entropy model (Sect. 3.3). The first procedure was applied to all the four cross-sections, whereas the latter was only applied to cross-sections 1 and 4, i.e., where the effects of the bridge piers are minimal so that the spanwise velocity distribution is unimodal.

## 3.1 Variability of the entropy function

Some relevant parameters that characterize the flow field (e.g., aspect ratio, average and maximum velocity) at the selected cross-sections are presented in Table 2 for the peak flow condition of the three flood events. The values of the entropic function, $\phi(M)_{\text{CFD}}$, were first computed as the ratio of average to maximum velocity within the cross-section provided by the 3D-CFD model. Then, assuming the site as ungauged, $\phi(M)_{\text{Eq (4)}}$ were estimated using Eq. (4) with $d_{90} = 0.01$ m (Pilbala et al., 2024) and a Manning parameter, $n$, equal to 0.035 m$^{-1/3}$s at the upstream ($-50$ m) and far downstream sections ($+100$ m and $+200$ m), and equal to 0.055 m$^{-1/3}$s just downstream of the bridge ($+50$ m cross-section) where larger energy losses are expected because of the wakes generated by the bridge piers. The values of $\phi(M)_{\text{Eq (4)}}$ reported in Table 2, corresponding with the points marked with dashed lines in Figure 3b, were obtained using $\xi = 5$ to compute $y_0$ (Sect. 2.5 just after Eq. (4)), and the grey band was obtained by varying $\xi$ in the range [1,10]. Finally, the values of the entropic parameter associated to the different values of $\phi(M)$ are computed using Eq. (2).

| Year | Distance from the bridge (m) | Channel aspect ratio (width/depth) | Average Velocity (m/s) | Maximum Velocity (m/s) | $\phi(M)_{\text{CFD}}$ | $M_{\text{CFD}}$ | $\phi(M)_{\text{Eq (4)}}$ | $M_{\text{Eq (4)}}$ |
|---|---|---|---|---|---|---|---|---|
| | $-50$ | 9.26 | 4.43 | 6.82 | 0.650 | 1.91 | 0.659 | 2.04 |
| 2012 | $+50$ | 13.78 | 2.91 | 7.01 | 0.415 | $-1.03$ | 0.410 | $-1.10$ |
| | $+100$ | 11.05 | 3.61 | 6.68 | 0.541 | 0.50 | 0.643 | 1.81 |
| | $+200$ | 8.5 | 4.06 | 5.48 | 0.740 | 3.4 | 0.642 | 1.80 |
| | $-50$ | 16.3 | 3.0 | 4.21 | 0.711 | 2.87 | 0.635 | 1.70 |
| 2019 | $+50$ | 18.45 | 1.93 | 3.74 | 0.515 | 0.18 | 0.405 | $-1.16$ |
| | $+100$ | 14.75 | 2.08 | 3.26 | 0.639 | 1.75 | 0.630 | 1.70 |
| | $+200$ | 12.84 | 2.40 | 3.47 | 0.690 | 2.51 | 0.609 | 1.35 |
| | $-50$ | 27.7 | 2.57 | 3.40 | 0.755 | 3.71 | 0.625 | 1.56 |
| 2022 | $+50$ | 27.3 | 1.33 | 2.58 | 0.514 | 0.17 | 0.395 | $-1.29$ |
| | $+100$ | 20.9 | 1.55 | 2.19 | 0.711 | 2.86 | 0.612 | 1.39 |
| | $+200$ | 13.23 | 1.97 | 2.56 | 0.767 | 3.96 | 0.619 | 1.48 |

**Table 2. Flow data for the cross-sections of Figure 2 and the three considered flood events of Table 1. The values of the entropic function, $\phi(M)$, and parameter, $M$, are obtained from the 3D-CFD velocity distributions and estimated according to Eq. (4).**

Since the entropic function is typically assumed to be constant for all flow conditions at a given cross-section, it is of interest to analyze its actual variation by exploiting the flow fields provided by the 3D-CFD model, and to see the effectiveness of their first-guess estimates obtained using Eq. (4). The values of $\phi(M)$ reported in Table 2 are plotted in Figure 3 as a function of the downstream distance from the bridge. At the first cross-section downstream of the bridge (i.e., cross-section 2), although referring to different flow conditions, values of the entropic function computed with the 3D-CFD and the current meter velocity distributions show the same magnitude, further confirming the reliability of the 3D-CFD model. The first-guess estimates of $\phi(M)$ in Figure 3b, although having a marginal role on the entropy-based computations, show a similar trend to the 3D-CFD

estimates (Figure 3b), provided that increased Manning parameter is used at the section just downstream of the bridge. The need to calibrate such an increased Manning parameter complicates efforts in case of disturbed flows.

For each flood event, at cross-sections 1 and 4, i.e., where the flow field is not characterized by the wakes generated by the bridge piers, the entropic function assumes similar values, which can be identified as "undisturbed" values. The variability of such undisturbed values of $\phi(M)$ with the flowrate is relatively small, as all the values fall in the range $0.65 < \phi(M) < 0.75$, which is in agreement with the range found by Bahmanpouri et al. (2022b) for similar European rivers. On the contrary, at cross-sections 2 and 3, just downstream of the bridge, the values of $\phi(M)$ are consistently reduced due to the effect of the bridge. At cross-section 2, for the flood conditions of 2012, $\phi(M)_{CFD}$ equals to 0.415, which leads to $M_{CFD} = -1.03$. The low value of $\phi(M)$ and the negative value of $M$ attest the strongly non-uniform distribution of the velocity (i.e., the maximum velocity in this cross-section is much higher than the average velocity). For the moderate peak flows of 2019 and 2022 event, the entropic function recovers undisturbed values already at cross-section 3, i.e., 100 m downstream of the bridge. In the largest flood event of 2012, which produced near pressure-flow conditions at the bridge with marked localized increasing of the flow velocity, $\phi(M)$ decreases from 0.64 to 0.42, and a sensible reduction is still present 100 m downstream of the bridge (cross-section 3).

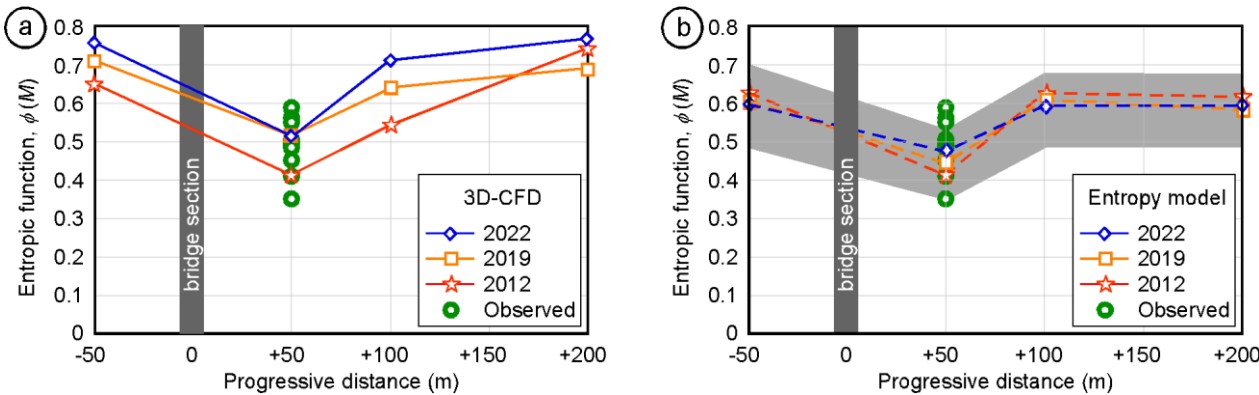

**Figure 3. Entropic function $\phi(M)$, for the different simulated scenarios, as a function of the distance from the bridge (positive downstream), (a) computed from the 3D-CFD flow fields and (b) estimated with Eq. (4), where the lines refer to the average values and the gray band is obtained by varying the reference height $y_0$ in Eq. (4) within the expected range. Green circles refer to data derived from velocity distributions measured with the current meter just downstream of the bridge.**

This first analysis suggests that assuming constant values of $\phi(M)$ can be reasonable in undisturbed river reaches; however, in case of irregular flow fields induced by the interactions with in-stream structures, the entropic function $\phi(M)$ can vary with respect to undisturbed values and, in addition, it can show significant variations with the flowrate.

### 3.2 Entropy model forced with the river-wide profile of free-surface velocity

The efficacy of the entropy model is here tested for the case in which the surface velocity is known for all the width of the cross-section. This could be the case in which the river-wide surface velocity is estimated from imaging techniques (e.g., Eltner

et al., 2020; Schweitzer and Cowen, 2021). The results, in terms of cross-sectional velocity distributions, are presented for brevity only for the intermediate peak flow of the 2019 flood event, and for the most challenging cross-sections just downstream of the bridge, where the flow field is disturbed by the pier wakes. The same results, for the peak flows of 2012 and 2022 events, are provided as supplementary material.

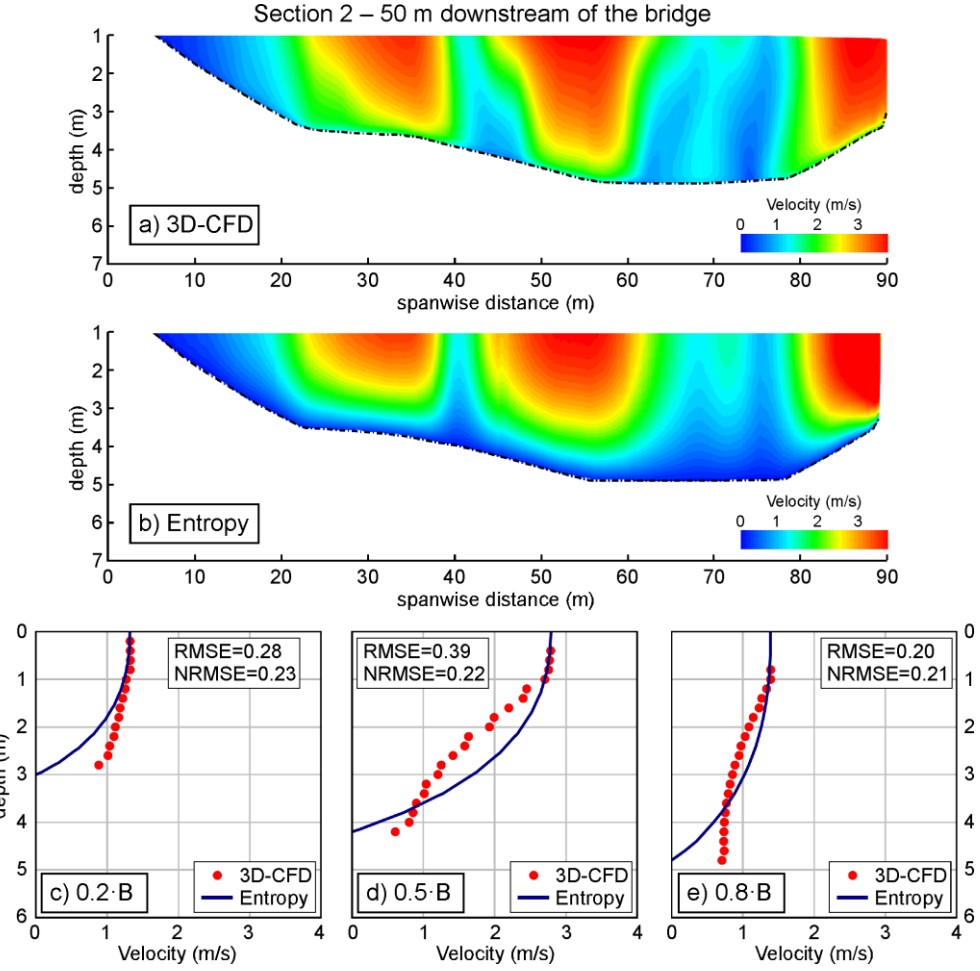

**Figure 4. Flood event of 2019, cross-section 2 (50 m downstream of the bridge). Velocity distributions provided by (a) the 3D-CFD model, and (b) the entropy model forced with the river-wide distribution of the free-surface velocity. Comparison of vertical distributions of velocity at 0.2B (c), 0.5B (d), and 0.8B (e), where B is the width of the cross-section.**

Figure 4 presents the cross-sectional velocity distribution 50 m downstream of the bridge (cross-section 2). As shown by the 3D-CFD flow field (Fig. 4a) and reflected in the low value of $\phi(M)$ for this cross-section (Table 2 and Figure 3), the effect of

the piers is very strong, such that there is a clearly uneven distribution of the cross-sectional velocity because of the wakes developing downstream the piers. Despite that, using as input the river-wide distribution of the surface velocity provided by the CFD simulation, the entropy model can reliably capture the salient features of the cross-sectional velocity distribution.

Figure 4(c-e) highlights the comparison of 3D-CFD and entropy flow velocities along three verticals located at 0.2 *B*, 0.5 *B*, and 0.8 *B* (where *B* is the channel width). Compared to the results of the 3D-CFD model, the entropy approach underestimates
the velocity close to the bed. Just downstream of the bridge, due to the presence of the bridge arches, the flow field provided by the 3D-CFD model is configured as a sort of partial orifice flow that increases the vertical uniformity of the velocity distribution compared to a uniform shear flow. Of course, the entropy model cannot capture such a localized flow features, which entails some difference in the patchiness of the physics-based and the entropy velocity distributions (Figure 4a-e). Since the velocities and the volumetric fluxes are still relatively small near the bed, these discrepancies marginally affect the
estimation of the section-averaged velocity and, consequently, of the total discharge (Table 3). The percentage error is quite larger (7.6%) for the very high-flow condition of the 2012 event (see Supplementary Materials), due to the accentuation of orifice-flow conditions associated to the higher water levels.

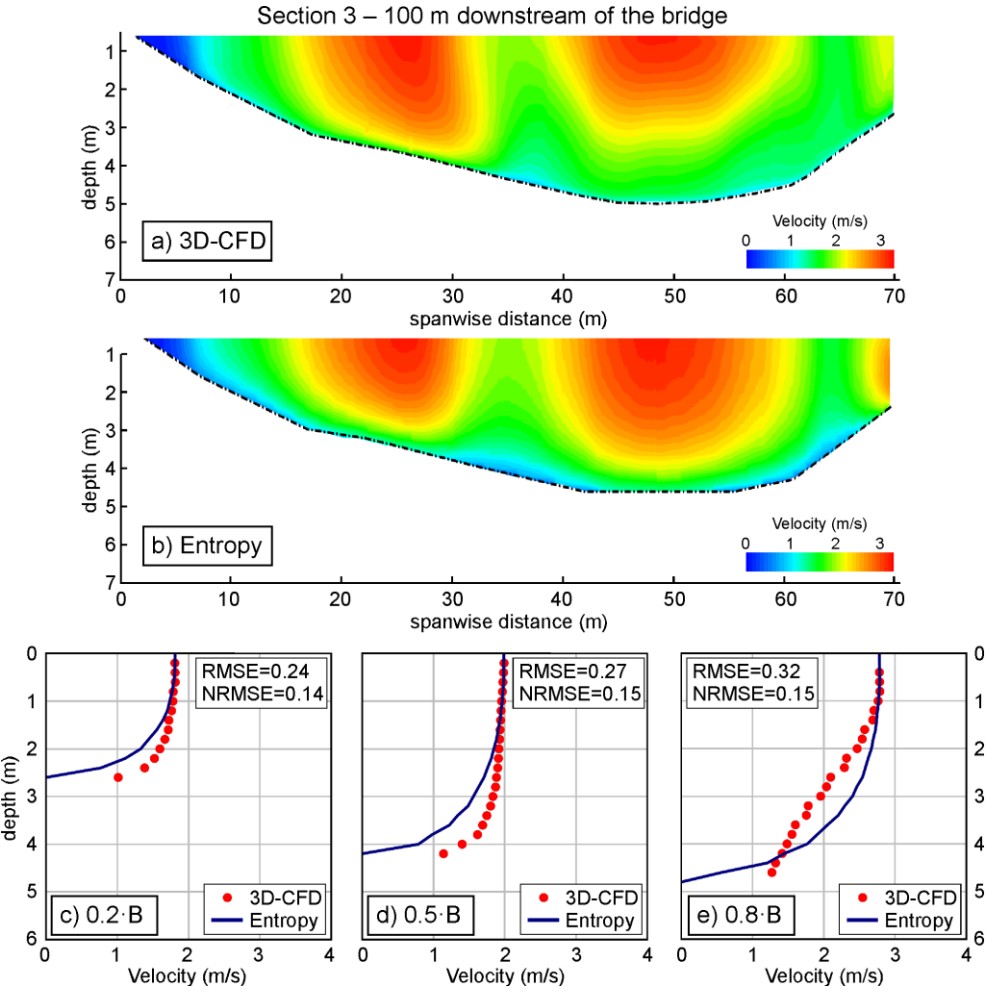

**Figure 5. Flood event of 2019, cross-section 3 (100 m downstream of the bridge). Velocity distributions provided by (a) the 3D-CFD**
**model, and (b) the entropy model forced with the river-wide distribution of the free-surface velocity. Comparison of vertical distributions of velocity at 0.2*B* (c), 0.5*B* (d), and 0.8*B* (e), where *B* is the width of the cross-section.**

Figure 5 depicts the cross-sectional velocity distributions at a larger distance from the bridge, i.e. at cross-section 3, placed 100 m downstream the bridge. The visual comparison with Figure 4 suggests that the effects of the piers on the flow field are reduced because of the increased distance, and the cross-sectional distribution provided by the 3D-CFD model (Figure 5a) appears more regular. The statistical analysis confirms that in this case the entropy model (Figure 5b) is able to simulate the velocity profiles with a higher accuracy.

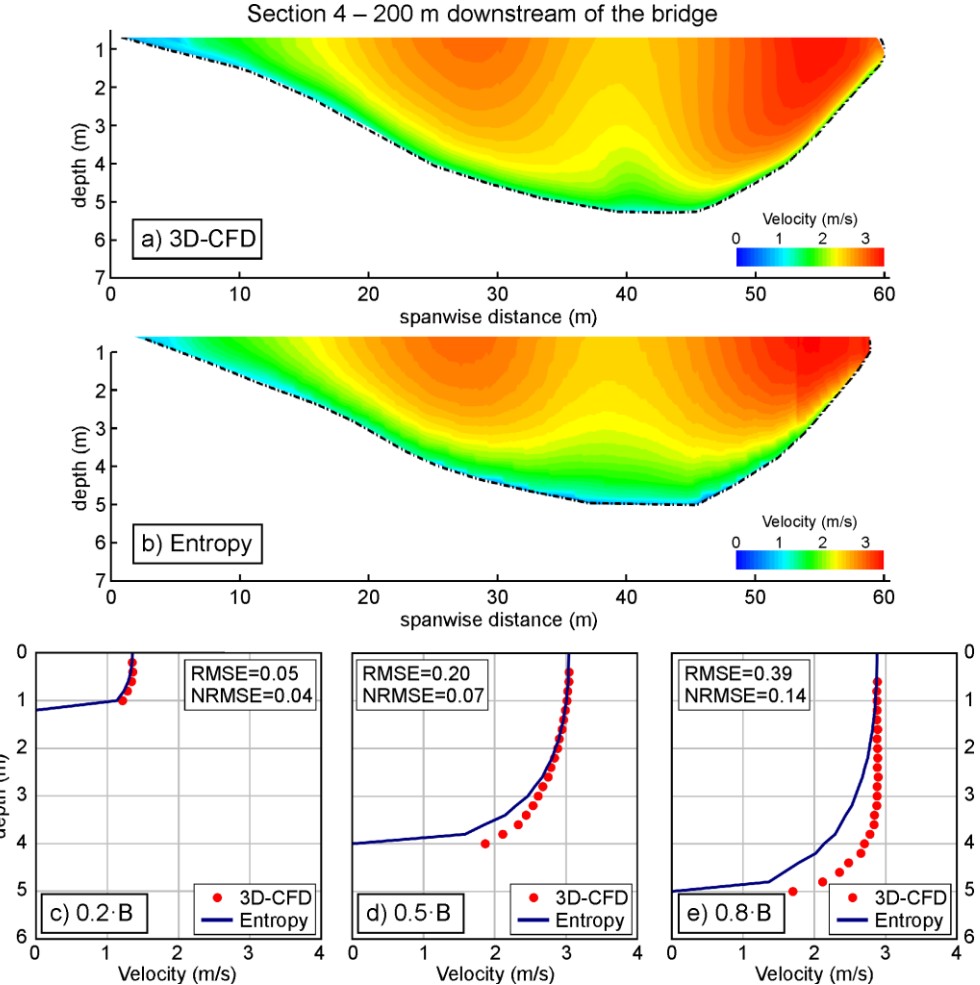

**Figure 6. Flood event of 2019, cross-section 4 (200 m downstream of the bridge). Velocity distributions provided by (a) the 3D-CFD model, and (b) the entropy model forced with the river-wide distribution of the free-surface velocity. Comparison of vertical distributions of velocity at 0.2$B$ (c), 0.5$B$ (d), and 0.8$B$ (e), where $B$ is the width of the cross-section.**

Figure 6 shows the cross-sectional velocity distributions of 3D-CFD and entropy models for the cross-section 4, i.e. 200 m downstream of the bridge. Compared to cross-section 3, the effect of the bridge piers is further reduced, because of both the distance and the more compact shape of the cross-section. Since the effect of the bridge piers is minimum, the statistical analysis shows a better agreement of the entropy model results with the CFD-based data. Though areas with high velocities

are still visible in simulations with higher values of the discharge (i.e., events of 2012 and 2019), for the high-flow conditions of 2022, the effect of the bridge pier has completely vanished. Therefore, the lower the flow discharge the lower the distance from the bridge to reach the normal flow condition without the bridge effect.

The results here presented show that, when the river-wide distribution of the free-surface velocity is provided, the entropy method allows to provide good estimations of the cross-sectional velocity distribution even when the influence of bridge piers,
and thus the unevenness of the flow field, is relevant. The main discrepancies are observed in the regions of flow with low values of velocity, which slightly affect the estimation of the flow discharge. Table 3 lists some statistics and error percentages for the depth-averaged velocity and discharge estimates for all cross-section and the three events considered. The estimation provided by the entropy method are in good agreement with results of CFD model, both upstream and downstream of the Adunata bridge. Though the accuracy is slightly reduced downstream of the bridge, the results are reliable also in the vicinity
of the structure (i.e., at cross-section 2), suggesting the applicability of the entropy model to estimate the flow discharges even in case of irregular distributions of the cross-sectional velocity, provided that the river-wide distribution of the surface velocity is used as input data.

| Flood event | Cross-section | Distance from the bridge (m) | Average velocity (m/s) | | Discharge (m³/s) | | Error percentage (%) |
|---|---|---|---|---|---|---|---|
| | | | 3D-CFD | Entropy | 3D-CFD | Entropy | |
| 2012 | 1 | −50 | 4.43 | 4.64 | 1'800 | 1'885 | +4.7 |
| | 2 | +50 | 2.91 | 2.69 | | 1'664 | −7.6 |
| | 3 | +100 | 3.61 | 3.54 | | 1'765 | −2.0 |
| | 4 | +200 | 4.06 | 4.30 | | 1'906 | +5.9 |
| 2019 | 1 | −50 | 3.0 | 3.0 | 450 | 450 | +0.1 |
| | 2 | +50 | 1.93 | 1.90 | | 443 | −1.5 |
| | 3 | +100 | 2.08 | 2.12 | | 459 | +2.0 |
| | 4 | +200 | 2.40 | 2.54 | | 476 | +5.8 |
| 2022 | 1 | −50 | 2.57 | 2.66 | 160 | 166 | +3.7 |
| | 2 | +50 | 1.33 | 1.24 | | 150 | −6.3 |
| | 3 | +100 | 1.55 | 1.51 | | 157 | −1.9 |
| | 4 | +200 | 1.97 | 1.98 | | 161 | +0.6 |

**Table 3. Comparison between 3D-CFD outputs and entropy-based estimations forced with the river-wide distribution of the free-**
**surface velocity.**

### 3.3 Entropy model forced with a single value of free-surface velocity

In this section, the results are presented considering only a single value of the surface velocity as input for the entropy model, which corresponds to the maximum surface velocity predicted by the 3D-CFD model. Two different spanwise velocity distributions are enforced in the entropic model, namely a parabolic spanwise distribution (PSD) and an elliptic spanwise
distribution (ESD). Of course, applying the entropy model using a unique value of the velocity is particularly sensitive of this value and supposes a unimodal velocity distribution in the spanwise direction. For this reason, this kind of approach cannot be

used in the cross-sections immediately downstream the bridge, where velocities show large spatial variations (see e.g., Figure 4). Herein, the results are presented for cross-section 1, i.e. 50 m upstream of the bridge for the high-flow condition of the 2012 event, and for cross-section 4, i.e. 200 m downstream of the bridge for the modest peak flow condition of the 2022 event,

where the effect of bridge piers on the velocity distribution wears off in a shorter distance.

Figure 7 shows the distribution of the surface velocity based on CFD outputs and both the PSD and ESD entropy models. The agreement of both the PSD and the ESD is generally good in the central and the right parts of the channel, and less good in the left part of the channel. Here, due to the irregular bathymetry (i.e., gravel deposit), the 3D-CFD model predicts localized stagnation zones that cannot be captured by the entropy model based on a single value of the surface velocity. This is confirmed

by Figure 8, which shows the cross-sectional distribution of the surface velocity and three vertical profiles. In the perspective of estimating the flow discharge, the lateral discrepancies represent a minor limit, as the central part of the cross-sections conveys the largest part of the total discharge.

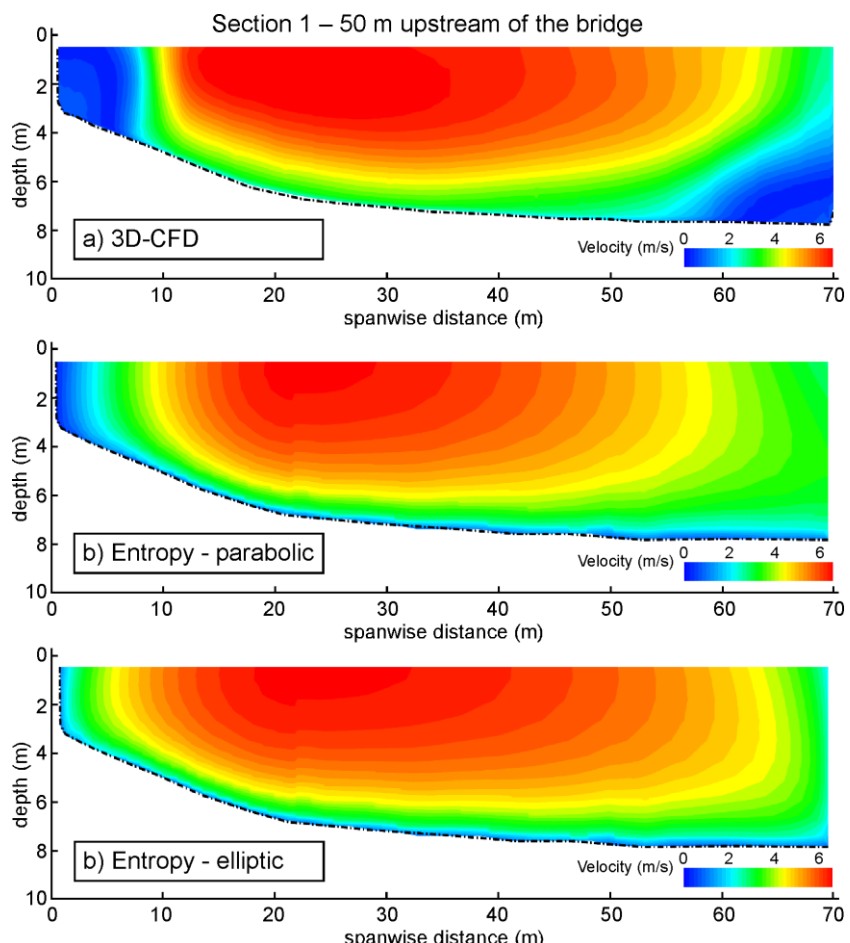

**Figure 7. Flood event of 2012, cross-section 1 (50 m upstream of the bridge). Cross-sectional velocity distribution computed with the**
**3D-CFD model (a), the entropy theory with parabolic (b) and elliptic (c) spanwise velocity distribution.**

Overall, the cross-sectional velocity distributions based on ESD seem more accurate than those based on the PSD: they provide similar results at the center of the channel, but the parabolic distribution generally underestimates the flow velocity close to the banks. Both cross-sectional and vertical distributions of the velocity profiles (Figure 7a and Figure 8c) highlight the existence of a velocity dip, i.e. the maximum velocity is below the water surface, particularly at the center of the channel. This

is generally the consequence of secondary currents superposed to the main flow (Termini and Moramarco, 2020). Yang et al. (2004) and Moramarco et al. (2017) reported that for large aspect ratios of channel flow, $B/D$, the dip phenomenon appears primarily near the sidewall region, whereas for relatively low aspect ratios ($B/D = 9.26$ for cross-section 1) the velocity dip is generally located at the center of the channel (Bahmanpouri et al., 2022a,b; Kundu and Ghoshal, 2018; Moramarco et al., 2017; Termini and Moramarco, 2020). In this case, the 3D flow field from the CFD simulation shows that the dip depends on the

counter-clockwise rotating secondary current generated by the upstream right-handed bend. Indeed, rotational inertia makes these curvature-induced helical flow structures to propagate downstream for relatively long distances (Dominguez Ruben et al., 2021; Lazzarin and Viero, 2023; Thorne et al., 1985).

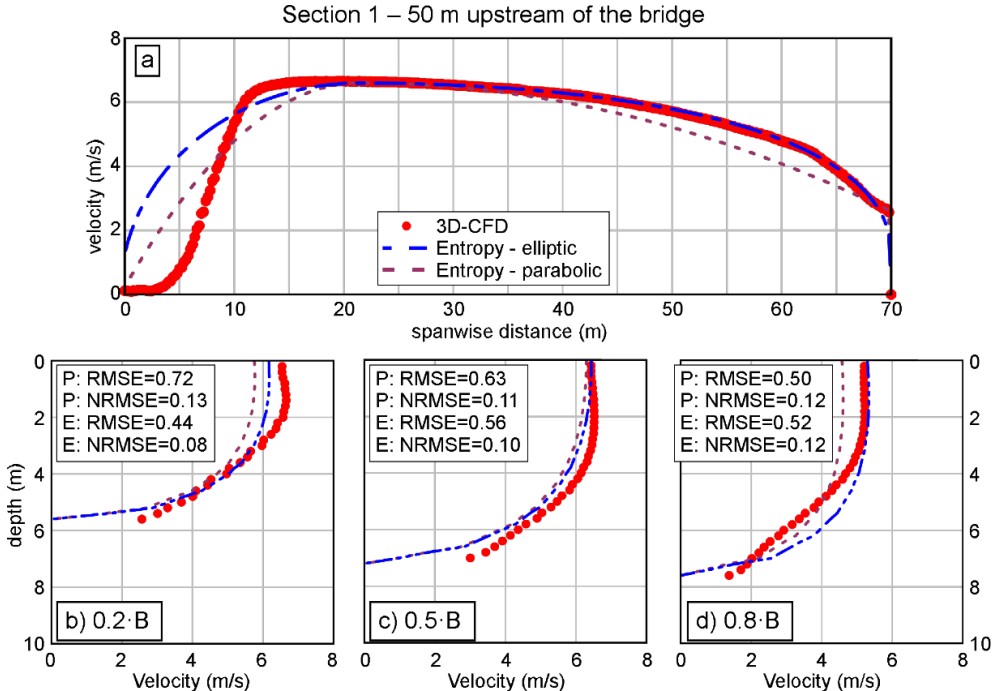

**Figure 8. Flood event of 2012, cross-section 1 (50 m upstream of the bridge). Spanwise distribution of the surface velocity (a);**
**comparison of vertical distributions of velocity at 0.2*B* (b), 0.5*B* (c), and 0.8*B* (d).**

The distribution of the velocity for the cross-section 4 (200 m downstream of the bridge) is presented in Figure 9 for the moderate peak flow condition of the 2022 event. For this cross-section, in the 3D-CFD results (Figure 9a), the maximum surface velocity is located on the left side of the channel, rather than at its center (this aspect is discussed in the following). Forced with the maximum water surface velocity, the entropy model well reproduces the velocity field in the central part of

the riverbed. Larger discrepancies are instead observed in the lateral part of the cross-section, with the elliptic spanwise distribution (ESD) that performs slightly better than the parabolic (PSD), particularly in the right side. Figure 10 shows the cross-sectional distribution of the surface velocity and the velocity distribution along three verticals. In terms of cross-sectional average velocity and flow discharge, both the PSD and ESD produce error that are lower than 10% (Table 4), then quite larger than those obtained using the river-wide surface velocity as input for the entropy model.

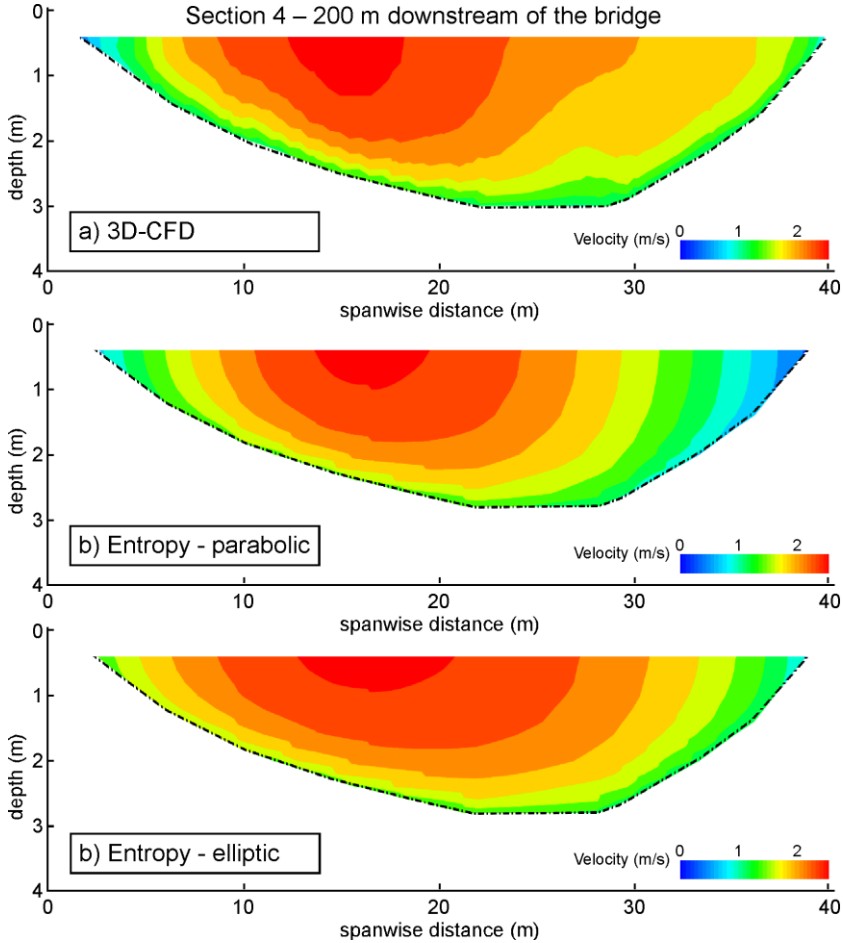

**Figure 9. Flood event of 2022, cross-section 4 (200 m downstream of the bridge). Cross-sectional velocity distribution computed with the 3D-CFD model (a), the entropy theory with parabolic (b) and elliptic (c) spanwise velocity distribution.**

A last point worth of discussing regards the unusual cross-sectional distribution of flow velocity at section 4 (Figure 9a). The reason why the 3D-CFD model locates the maximum velocity at the left of the talweg is the alternate vortex shedding occurring downstream of the bridge piers, which propagates beyond the last considered cross-section. This is evident in the map of instantaneous surface-velocity of Figure 11. This particular occurrence poses interesting questions on the application of the entropy model to estimate the flow discharge downstream of in-stream structures. First, the spanwise location of the maximum surface velocity is subject to a periodical shift, which prevents its correct detection by means of a fixed sensor with a small-

size field of view, like the one mounted on the Adunata bridge. Secondly, marked time-varying flow fields, which occasionally
(or periodically) deviate from nearly uniform flow conditions, can hardly be captured by any preset velocity distribution. To
alleviate the problem, the periodic signal of surface velocity can be filtered, which is equivalent to look at time-averaged
modelled flow fields, which requires knowing the frequency of vortex shedding.

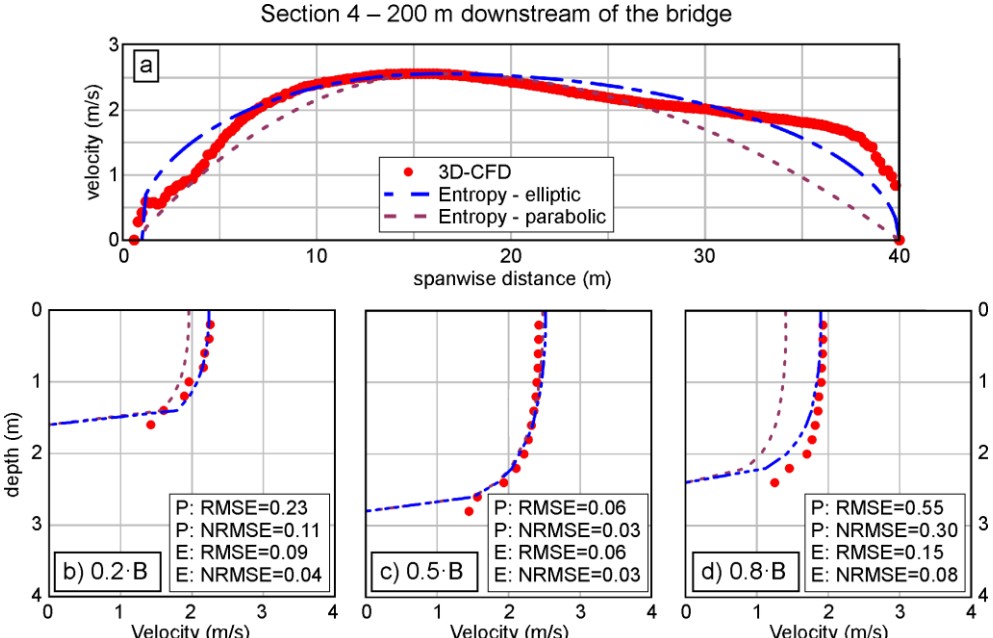

**Figure 10. Flood event of 2022, cross-section 4 (200 m downstream of the bridge). Spanwise distribution of the surface velocity (a);**
**comparison of vertical distributions of velocity at 0.2B (b), 0.5B (c), and 0.8B (d).**

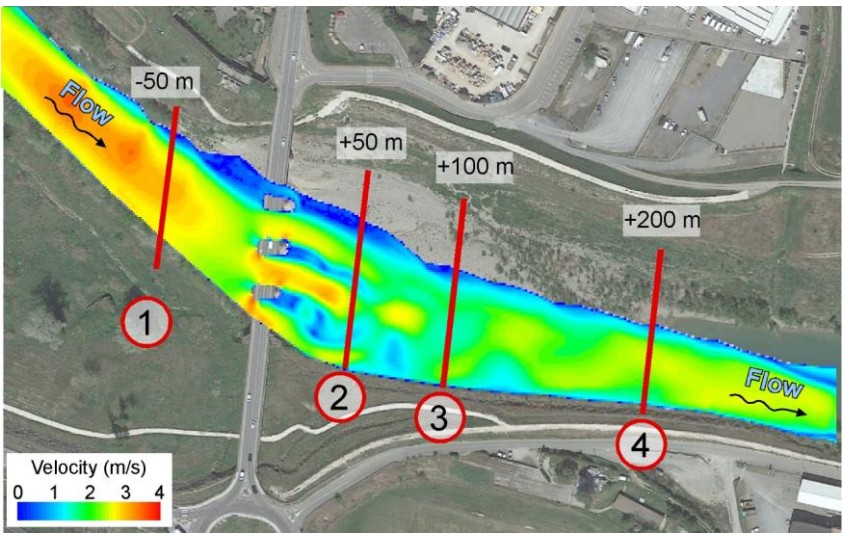

**Figure 11. Flood event of 2022. Colormap of the instantaneous surface velocities computed with the 3D-CFD model for the Paglia River at the Adunata bridge (aerial image from © Google Earth).**

The results shown in this Section confirm the general accuracy of the entropy model in predicting the velocity distributions.
As expected, when using a single value of velocity in place of the river-wide distribution of surface velocity, the accuracy of the method decreases. Provided that using a single velocity is beyond the scope of the method when the velocity distribution is markedly irregular, the entropy approach can still be forced with a single surface velocity, and produce accurate results, when there are no evidences of strong disturbances of the flow. Indeed, such an approach cannot capture marked unevenness in the flow field, as shown in the case of the lateral low-velocity regions at cross-section 1 for the 2012 event (Figure 7), and in the time-varying flow field of cross-section 4 for the 2022 event (Figure 9).

| Distance from the bridge (m) and year | Average velocity (m/s) | | | Discharge (m³/s) | | | Error percentage (%) | |
|---|---|---|---|---|---|---|---|---|
| | 3D-CFD | Entropy | | 3D-CFD | Entropy | | | |
| | | Parabolic | Elliptic | | Parabolic | Elliptic | Parabolic | Elliptic |
| −50 (2012) | 4.43 | 4.44 | 4.83 | 1'800 | 1'804 | 1'962 | +0.2 | +9.0 |
| −50 (2019) | 3.00 | 3.24 | 3.40 | 450 | 486 | 510 | +8.0 | +13.3 |
| −50 (2022) | 2.55 | 2.65 | 2.74 | 160 | 166 | 172 | +3.8 | +7.5 |
| +200 (2022) | 1.97 | 1.81 | 2.02 | 160 | 147 | 164 | −8.0 | +2.0 |

**Table 4. Comparison between 3D-CFD and entropy-based outputs considering a single surface velocity.**

## 4 Conclusions

The present study investigated the ability of the entropy-based method to estimate the cross-sectional distribution of velocity, as well as the associated river discharge, for different flow conditions in a representative case study. As sensors for continuous monitoring of water level and surface velocity are often mounted on existing bridges, a stretch of the Paglia River was analyzed where a multi-arch bridge with thick piers, which hosts a level gauge and a radar sensor, strongly affects the flow field.

With the goal of assessing the applicability of the entropy model in case of flows disturbed by the presence of in-stream structures, a 3D-CFD model was set up to obtain reliable, physics-based velocity distributions at relevant cross-sections, both upstream and downstream of the bridge. The entropy model was then applied to reproduce this set of velocity distributions, using the bathymetric data and the CFD-computed surface velocity as input.

As a first point, the study highlighted the potential of using accurate, physics-based, 3D-CFD models to deepen the knowledge of rivers and, specifically, of theoretical methods for discharge estimation. Indeed, 3D-CFD models allows providing pictures of complex flow fields that are more complete than, e.g., ADCP measures, in terms of spatial and temporal distribution and, above all, valid for high-flow regimes which typically prevent any direct measurement of the flow field beneath the free-surface. This entails unexplored chances of outlining best-practices in the use of simplified methods for continuous discharge monitoring, and, as a consequence, to improve their accuracy.

According to the present analysis, the entropy model revealed remarkable skills in reproducing also disturbed and uneven flow fields when the river-wide distribution of the surface velocity is used as input data. This occurred also just downstream of the bridge, where the pier-induced wakes made the velocity distribution multimodal and extremely irregular, with error on

discharge estimates lower than 8%. The availability of innovative measuring techniques, able to collect river-wide surface velocity data at a relatively low cost, adds value to the present findings.

On the other side, the accuracy of the entropy model is reduced when only the maximum surface velocity is used as input data, so that the spanwise velocity distribution has to be assumed on a theoretical basis (e.g., parabolic or elliptical). While such a method is absolutely discouraged in case of disturbed flow fields (e.g., downstream of in-stream structures), it still provides accurate estimates where the velocity field is sufficiently regular.

As a final recommendation, measuring instruments and sensors for surface velocity become more effective when placed upstream of in-stream structures, i.e., where the flow field is only marginally affected by the structure and both the water surface elevation and the velocity distribution are far more regular.

A main limitation of the present methodological approach relies in the assumption of fixed bed in both the CFD analysis and the application of the entropic model. In natural rivers, bed scouring during severe flood events and the ensuing formation of local deposits, especially close to in-stream structures such as bridges, can alter the bathymetry and, in turn, the velocity distribution and the discharge estimates. In case of movable bed and absence of protection measures (e.g., riprap or bed sills), the uncertainty associated to the local bed mobility has to be evaluated with due care. Future research on more complex scenarios that still need a comprehensive assessment, and which could largely benefit from physics-based numerical modelling, will include the case of mobile beds and the analysis of stage-dependent variations of cross-sectional velocity distribution, particularly in case of compound cross-sections that are typical of natural rivers.

## Appendix A

To impose the boundary conditions to the 3D-CFD model, a 2D depth-averaged model of a longer stretch of the Paglia River has been setup. We used the 2DEF Finite Element model (Defina, 2003; Lazzarin et al., 2023a, 2024; Viero, 2019; Viero et al., 2013, 2014), which solves a modified version of the shallow water equations (SWEs) that allow for a robust treatment of wetting and drying over irregular topographies (D'Alpaos and Defina, 2007; Defina, 2000). The SWEs are written as:

$$
\begin{aligned}
&\eta(h_s)\frac{\partial h_s}{\partial t} + \nabla \cdot \mathbf{q} = 0 \\
&g\nabla h_s + \frac{\mathrm{D}}{\mathrm{D}t}\left(\frac{\mathbf{q}}{Y}\right) + \frac{\boldsymbol{\tau}}{\rho Y} - \nabla \cdot \mathbf{Re} = 0
\end{aligned}
\tag{A1}
$$

in which $h_s$ is the free surface elevation, $t$ is the time, $\nabla$ and $\nabla \cdot$ denote the 2D gradient and divergence operators, respectively, $\mathbf{q} = (q_x; q_y)$ is the depth-integrated velocity (i.e., the unit-width discharge), $Y$ is the equivalent water depth (i.e., the volume of water per unit area), and $\eta(h_s)$ a storativity coefficient to account for the wetted fraction of the domain, $\boldsymbol{\tau} = (\tau_x; \tau_y)$ is the bed shear stress, evaluated using the Gauckler-Strickler formula, $\rho$ is the water density, and $\mathbf{Re}$ the horizontal components of the Reynolds stresses, modelled according to the Boussinesq approximation. A mixed Eulerian-Lagrangian approach allows evaluating the total derivative of the flow velocity in the momentum equations using finite differences and a backward tracing

technique based on the method of characteristics (Defina, 2003; Giraldo, 2003; Walters and Casulli, 1998). Then, the SWEs are solved with a finite element method, based on triangular, unstructured grids. The model also allows to couple 2D triangular elements with 1D elements (either open- or closed-sections) to model the minor hydraulic network efficiently; other 1D elements are used to model particular devices, such as pumps, weirs, etc. (Martini et al., 2004). The model has been successfully used to simulate flows in various rivers (e.g., Mel et al., 2020a,b; Viero et al., 2019; Baldasso et al., 2023); its effectiveness have been demonstrated also in different research field, such as lagoon and marine environments (e.g., Carniello et al., 2012; Pivato et al., 2020; Tognin et al., 2022; Viero and Defina, 2016).

In the present case, the computational mesh covered a stretch of the Paglia River about 7 km long, from 600 m upstream of the Adunata Bridge to the confluence with the Tiber River, including floodable floodplains (Fig. A1). The average mesh size ranged from 10 m in the riverbed near the Adunata bridge, to 30 m in the floodplains and far downstream of the Adunata bridge. The computational mesh included 61'000 triangular elements, 16 1D elements to simulate underpasses, and 4 1D weir elements to simulate the sill located 500 m downstream of the Adunata Bridge.

The inflow hydrographs, prescribed at the upstream mesh inlet, were derived from water levels measured at the Adunata Bridge using the associated rating curve. At the outlet, an arbitrary rating curve was applied as downstream boundary condition; a sensitivity analysis showed that, because of the distance from the Adunata Bridge, this boundary condition did not produce any perceivable effect in the water levels simulated at the study site.

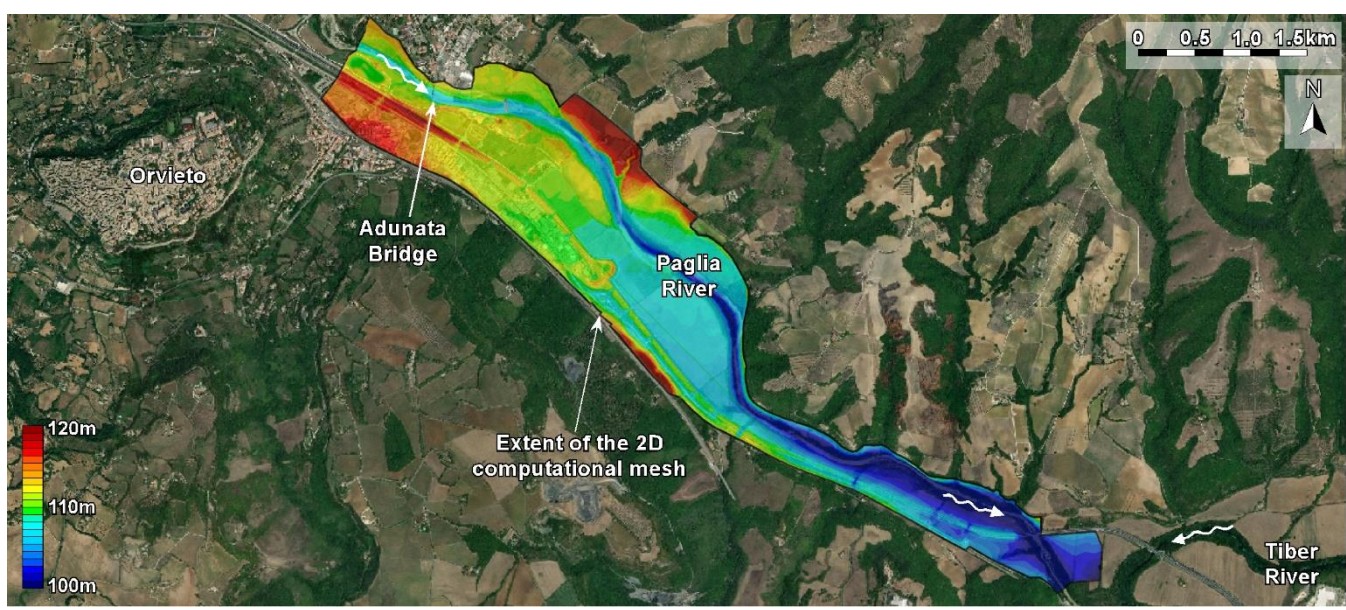

**Figure A1. Spatial extent of the 2D computational mesh (aerial image from World Imagery). The color map shows the bottom elevation of the grid elements derived from the LiDAR-based DTM.**

Different Gauckler-Strickler coefficients were assigned to the different parts of the domain (e.g., floodplains, densely vegetated areas, etc.) based on the soil cover. The value assigned to the main riverbed were calibrated to match the time series of the

490 water levels measured at the Adunata bridge gauging station for the 2019 flood event (Fig. A2a) and, for the most sever flood event occurred in 2012, the model results were also checked in terms of extent of flooded areas. The minor flood of 2022 was used to verify the model (Fig. A2b). Finally, the depth-averaged velocity just downstream of the Adunata Bridge was compared with the free-surface velocity measured by the radar sensor. Due to the use of a coarse grid and to the depth-average assumption, the 2D model underpredicted the measured water surface systematically (Fig. A2c,d); however, using an

495 amplification factor of 1.7 (gray dots in Fig. A2c,d), the predicted values were quite similar to the measured ones.

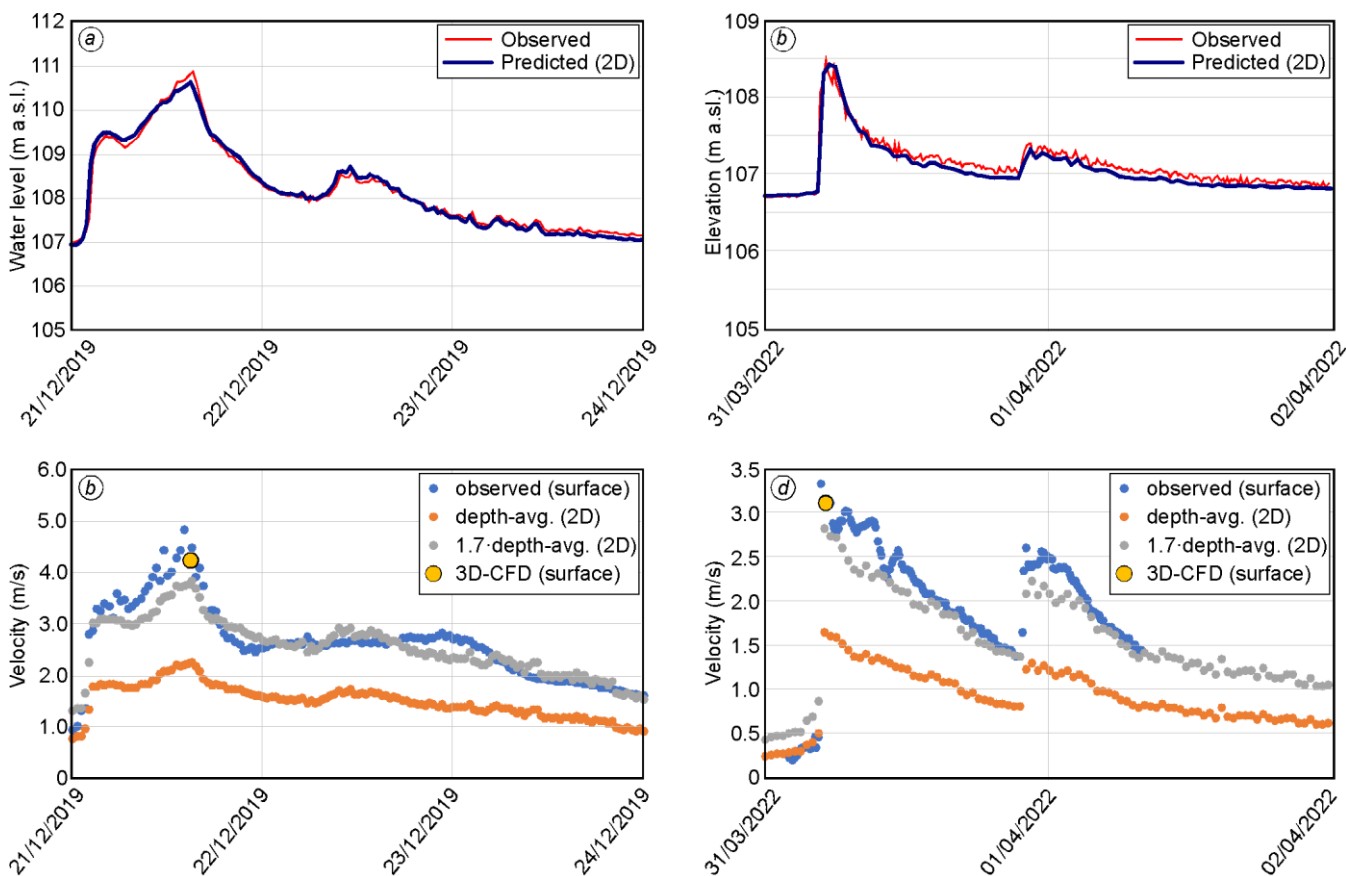

**Figure A2. Observed (red) and predicted (blue) water levels at the Adunata Bridge gauging station for the flood events of 2019 (a) and 2022 (b). Observed and predicted water velocity for the flood events of 2019 (c) and 2022 (d).**

**Data availability**

Data available on request from the authors.

### Author contribution

Conceptualization: FB, TL, SB, TM, DPV. Formal analysis: FB and TL. Funding acquisition: TM. Investigation: FB and TL. Methodology: FB, TL, SB, TM, DPV. Project administration: SB, TM, DPV. Software: FB, TL, TM, DPV. Supervision: SB, TM, DPV. Visualization FB, TL, DPV. Writing – original draft preparation: FB. Writing – review & editing: TL, SB, TM, DPV.

### Competing interests

The authors declare that they have no conflict of interest.

### Acknowledgements

This study was supported by Italian National Research Programme PRIN 2017, with the project n. 2017SEB7Z8 "IntEractions between hydrodyNamics and bioTic communities in fluvial Ecosystems: advancement in the knowledge and undeRstanding of PRocesses and ecosystem sustainability by the development of novel technologieS with fIeld monitoriNg and laboratory testing (ENTERPRISING)". T.L. is sponsored by a scholarship provided by the CARIPARO foundation. The authors acknowledge the assistance of Luigi di Micco, Shiva Rezazadeh, and Marco Dionigi.

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
