# Peer review of "Estimating velocity distribution and flood discharge at river bridges using the entropy theory. Insights from Computational Fluid Dynamics flow fields"

_Hydrology and Earth System Sciences, 2023_

## Referee Comment (RC2)

**General Comments:**

        The manuscript evaluates the efficacy of the two available approaches of estimating the velocity distribution of a given discharge passing at the Paglia River reach flow sections in the vicinity of the Adunata bridge subject to the conditions of with and without the impact of the presence of bridge piers on the velocity distributions of the flow estimated at these sections using the entropy theory. These velocity distributions are compared with that of the velocity distributions of the same discharge obtained at these respective flow sections based on the flow fields simulated by a Computational Fluid Dynamics (CFD) model, considered as the benchmark model. The CFD model is set up using different types of observed input data measured and collected based on the recorded water level, velocity measurements made using current meter, rating curves and free surface velocity data collected using the water level and velocity radar sensors mounted on the downstream of the Adunata bridge deck for the three considered past flow events of 2012, 2019 and 2022. Further, the velocity distributions estimated at the studied river flow sections based on the application of entropy theory were assessed using the ADCP measurements made at the section far away upstream of the Adunata bridge. To simulate the velocity distributions at the studied river sections using the entropy theory, the authors have adopted two approaches of using the surface velocities estimated by the CFD model simulations at the considered sections of the considered flow event, *viz.*, the use of span-wise simulated surface velocity estimates at the considered river section, and use of only the simulated maximum surface velocity estimate at the same section. Based on the study, the authors arrive at the conclusion that the span-wise estimated surface velocity measurements are needed to effectively capture the sectional velocity distributions close to that simulated by the CFD model at the sections immediately downstream of the bridge where the flow fields are impacted by the disturbance generated by the presence of bridge piers; whereas, the flow sections at far away upstream and downstream of the bridge, where the flow fields of the passing discharge are not impacted by the bridge piers, the velocity distributions estimated at these section using the entropy theory based only on the use of maximum surface velocity information may be sufficient for closely reproducing the CFD model based velocity distributions at these sections.

The study is timely and a needed one to widen the knowledge on the field applicability of the entropy theory for discharge estimation. It can be inferred from the study that when the flow field in a river reach is not impacted by the presence of a structure constructed across a river, then the measurement of maximum surface velocity may be sufficient to estimate the discharge passing at that section using the entropy theory. However, when the river section flow characteristics are impacted by the presence of a structure constructed across the river, then span-wise surface velocity measurements may be required to simulate the actual velocity distribution that prevails at that section which is required for serving the purpose of studying scour around the bridge piers. Therefore, both these different approaches of estimating surface velocity measurements using velocity radar(s) have their relevance to serve their intended practical purposes. The manuscript deserves to be accepted for the stated reasons. However, the authors need to address many comments and incorporate corrections in the manuscript, as given in the following pages, before its publication in the final form.

**Specific Comments:**

Comment-1: Since the manuscript describes the study carried out by the authors and then reported here, it would be appropriate to describe the study in the past tense rather than in the present tense, throughout the manuscript.

Comment-2: Since the main emphasis of the study is related to the velocity distributions of a given discharge passing through many flow sections of the river reach in the vicinity of a bridge,

the title of the manuscript may reflect on this aspect, specifically changed as

"Estimating the velocity distribution and the discharge passing at different flow sections of a river reach in the vicinity of a bridge using the entropy theory: Insights from the flow fields generated by a computational Fluid Dynamics model."

Comment 3: in Line #38, Explain, what is the secondary current of the second kind?

Comment 4: in Line #65, use of some field data! Which field data?

Comment 5: in Line #76, What do you mean by "weak gauging sites?" or is it wake affected gauging sites?

Comment 6: in Line #89, two European rivers! Specify these two rivers.

Comment 7: in Lines #106-107, severe flooding and high sediment transport! What is the impact of the high sediment rate transport on the flow velocity and its distribution under different magnitudes of sediment laden flood discharge?

Comment 8: in Line #117, "mayo axis", is it major axis?

Comment 9: in Line #167, it is stated that there different steady flow conditions are simulated using the 3D-CFD model which correspond to the peak flow conditions of flood events occurred in 2012, 2019 and 2022. But in Line #173, it is stated that surface velocity data are not available for the 2012 flood event. So how the 2012 flow event's velocity distribution was simulated using the CFD model for the peak discharge of the 2012 flow event?

Comment 10: in Line #185, define phi(M)!

Comment 11: in Line #208, Phi(M) is defined as entropy parameter, and in Line# 74, the parameter M is also defined as entropy parameter. So a consistent definitions of these parameters need to be given.

Comment 12: in Table-2 contents, the second line inside the Table the estimate of M is given as -1.03, what is the physical meaning of a negative M?

---

## Author Comment (AC1)

Reply to Comments by:

**REVIEWER 1**

The manuscript presents an application of entropy theory to determine the discharge at a river section from measurements of the surface velocity. The key features of the manuscript are (i) comparing the discharge estimates when a transverse profile of surface velocity is entirely known and when only a single-point value of surface velocity is known and (ii) considering sections where the flow distribution is significantly disturbed by interfering structures like bridges.

The manuscript is quite interesting but I feel that some revision could strengthen it and make the method more appealing for readers. I have some major concerns with this manuscript:

RE: We thank the reviewer for appreciating the paper and for the constructive criticism and comments, which allow us to improve the paper.

MAJOR COMMENTS

(1) Section 2.5, entitled "Entropy theory", provides some equations but no concept at all. Which is the grounding principle of this method? Which is the physical meaning of M and phi(M) or, in other words, what do these parameters parameterize? No need to rewrite referenced papers, obviously, but providing some conceptual ground would be needed to understand the description and interpret later findings (for example, at line 234 it is mentioned that the disturbance by the bridge reduces the entropy parameter; why? Which is the physical interpretation?).

RE: We agree about the need of providing some basic concepts of the entropy model and of clarifying the physical concepts of the involved parameters. We enriched the text of Sect. 2.5 with some basic concepts on the entropy theory, the physical meaning of the entropy function, $\phi(M)$, which is the ratio of the average to maximum flow velocity, of the associated entropy parameter, $M$, and their typical behaviour in rivers as outlined by previous studies. For example, the disturbances produced by the bridge act to increase the maximum velocity with respect to its average value, i.e., to reduce the entropic function $\phi(M)$ which is defined as the ratio of average to maximum velocity in the cross-section. This is coherent with the fact the localized geometrical variations can produce local accelerations and increase the unevenness of the cross-sectional velocity distribution.

(2) The authors should better describe how they use data and to do what. My understanding is that they rely on surface velocity measurements by ADCP and on the results of the 3D simulations. The ADCP measurements are used to "validate the numerical models" (line 136), while the entropy theory is applied to the simulated flow fields. For the latter, the procedure is: (i) determining the entropy parameters phi(M) and M based on the simulated velocity distribution in a cross section (obtaining the values listed in Tab 2 and plotted in Fig 3); (ii) determining a velocity distribution in the cross section based on the surface velocity from the simulations (entire profile or single-point); (iii) integrating the obtained velocity distribution; and (iv) compare the discharge with that used as an upstream boundary condition in the simulations. Now, steps (i) and (ii) sound like creating a loop, more or less like when the same data are used for calibration/training and validation. It is not completely so, since a bulk estimate of an entropy parameter is later used to determine a velocity profile at every vertical, but some self-dependence should be present. Can the authors comment on this issue? Or, I may have misunderstood the procedure, that would need more explanation.

RE: We admit that the use of different data was not clearly described in the paper, and we apologize for that. We clarified the point at the end of Section 2.5 (and in other part of the text) by better explaining how the entropic model is applied, which data is used as input and which data only to check the accuracy of the outputs. We are confident that this allows to better understand the use of data and that there are no loops in the adopted procedure.

Basically, the entropic distribution of the cross-sectional velocity depends the maximum velocity (for each vertical or for the whole cross-section), on the depth at which the maximum velocity occurs (i.e., the so-called dip), and on the entropic parameter, $M$, which has a one-to-one relationship with the entropic function, $\phi(M)$. The entropic function, $\phi(M)$, has a clear physical meaning, as it is the ratio of the average to maximum velocity in the cross-section. In our analysis, we first computed $\phi(M)_{CFD}$ from the cross-sectional velocity distributions that have been measured and simulated with the 3D-CFD model, to check how the physics-based estimates of $\phi(M)$ vary in the considered cross-sections. To apply the entropy model, $\phi(M)$ and $M$ were estimated using literature formulae and used in an iterative procedure that is independent of the values $\phi(M)_{CFD}$ obtained from the 3D-CFD model, which avoided creating a loop. Indeed, of the flow fields computed by the 3D-CFD model, we used the surface velocity at the four cross-sections (either their river-wide distribution or only the maximum value) as input data for the entropy model, and then we used the 3D-CFD cross-sectional velocity distributions as a benchmark for the velocity distributions computed with the entropy models. We admit that these aspects were not clearly explained in the previous text.

(3) It is not clear to me how this method could be applied at a section where the entropy parameter is not known, starting just from a profile of surface velocity. My impression is that, in such a case, a tentative value of the entropy parameter should be used. How estimated? Expert judgement leading to sound values, as we normally do for roughness? I note that also the iterative procedure described at lines 192 and following "can be applied for sites with a given phi(M)". Related to this issue, I note that an estimated discharge sounds quite sensitive to the entropy parameter, as the two quantities are linearly dependent based on eq. (2). The problem is the same that affects discharge determination based on large-scale PIV (e.g., 10.1029/2008WR006950, 10.1016/j.jhydrol.2010.05.049) where a coefficient is used to pass from surface to depth-averaged velocity. The coefficient is not known, but most references use 0.85 as a suitable value. In the present context, one might use phi(M) = 0.72 for "undisturbed" flows (average of values for -50 and +200 in Tab 2), but it will be hard to have a robust indication for disturbed flows.

RE: The issues raised in this comment are indeed related to the unclear aspects mentioned in the previous comments. We admit that the procedure used to apply the entropy model was not described with due clarity. The sentence "can be applied for sites with a given phi(M)" was wrong as de dip, $h$, is computed based on formula (now added to the text) derive in laboratory experiments that does not depend on phi(M). However, for cross-sections in which only the surface velocity is known (and there is not previous information on the cross-sectional velocity distribution), an iterative method is used in which the first guess for $\phi(M)$, computed with a standard formula, allows determining the first-iteration velocity distribution. Then, based on this cross-sectional velocity distribution, $\phi(M)$ and, in turn, $M$, are recomputed. The procedure is repeated until the difference between successive values of $\phi(M)$ becomes less than a small threshold value, e.g., 0.01. Moramarco et al. (2017) demonstrated that the vertical distribution of the velocity based on the Entropy method is generally more accurate than that estimated based on the velocity index method. Furthermore, while the velocity index model gives only the average velocity, the entropy approach estimates the whole cross-sectional distribution of the flow velocity, which in turn allows estimating the other quantities of interest, such as the bed shear stress, etc.

(4) I would suggest to carefully reconsider the use of "at" river bridges in the title. My initial interpretation was that the focus of the paper would be on sections immediately upstream or downstream of the bridge, and I think that most hydraulicians would have done the same. Instead, the paper considers sections in the vicinity of the bridge but at some distance from it (some/several widths). This is quite important for applying the method using monitoring data since, as remarked by the authors, instrumentation is generally mounted on the bridge.

RE: We understand the comment and acknowledge that the cross-section we used in the analysis are not located exactly at the bridge. The sections that are closest to the bridge are 50 m upstream and downstream of the bridge, a distance which is about $0.45B$, with $B$ the width of the river at the bridge section. These sections are indeed very close to the bridge, particularly considering that, while the water level sensors measure perpendicularly, the remote sensors for surface velocity (such as radar, Large Scale PIV, etc.) have their field of view located some tens of meter upstream or downstream of the bridge. Similarly, when a current meter is operated from a bridge, the drag force moves the current meter downstream at a distance that depends on the height of the bridge with respect to the free-surface elevation and on the drag force itself. Finally, note that the flow fields predicted by the CFD model show that the velocity distribution varies abruptly at the upstream edge of the bridge (flow contraction), and smoothly in the tens of meter upstream and downstream of the bridge. Accordingly, for the purpose of the study, we remain convinced that the particular choice of looking at cross-sections located 50 m upstream and downstream of the bridge can still be regarded as sections "at" the bridge. We are reluctant to use "in the vicinity" because the title is already quite long (and it was extended to include "velocity distribution" according to the suggestion by Reviewer #2). We added some text at the beginning of Sect. 3 to explain this issue:

> "The sections just upstream and downstream of the bridge are located at a distance of about $0.45B$ from the bridge, with $B$ the width of the river at the bridge section. This is a short distance, particularly considering that the remote sensors for surface velocity (such as radar, Large Scale PIV, etc.) have their field of view located some tens of meter upstream or downstream of the bridge. The sections far downstream are considered to assess how far the flow field is affected by the presence of the bridge."

Finally, the fact that we also considered two cross-sections further downstream of the bridge is only meant to assess how the disturbances generated by the bridge vary in space. The core of the study is an attempt to answer the question: "how can we estimate the discharge with the entropic theory and measures carried out from river bridges?", which we believe is well reflected by the modified title.

MINOR COMMENTS (listed by line)

16: I think that mentioning explicitly that 12 sample applications are considered (3 flow rates by 4 sections) would make the manuscript findings sound more robust.

RE: We agree. The Abstract was modified accordingly:

> "A total of 12 samples, including three different flow conditions for four cross-sections, one upstream and three downstream of the bridge, are considered. "

17: not sure that the word "safely" is the appropriate one to state that the method is satisfactorily accurate.

RE: We agree. We changed "safely" with "reliably".

50-68: I find these lines a bit hard to follow, due to the lack of a clear purpose. To me, the point is that one would like to determine Q based on its definition, that is integral(v)dA. Since the cross-sectional distribution of velocity is unknown, a method to infer the cross-sectional velocity distribution from other data (that are in the end surface velocities) is sought. I would recommend to use this line of thought to revise these lines.

RE: Thank you for the feedback. We reworded and re-organized most parts of the Introduction, that was actually a bit confused.

70: the method based on a ratio between surface and depth-averaged velocities could be mentioned as an alternative one (some references are found in major comment 3).

RE: We mentioned the method indicated by the reviewer indicating the advantages of the proposed entropy approach. Now the text reads:

> "One advantage of the entropy approach is providing the complete cross-sectional distribution of velocity. By contrast, other indirect methods for estimating flow discharge only compute the depth-averaged value from the surface velocities at subsections using a fixed reduction coefficient (e.g., Le Coz et al., 2010)."

76: what is meant with "weak" gauging site?

RE: We intended a gauging site with insufficient data to obtain reliable estimates of the flow discharge. This part of the text has been deleted to improve the readability of the Introduction.

107: if the river is characterized by high sediment transport, one can expect relevant morphologic changes. The simulations carried on in this work are with a fixed bed and always the same geometry; this is not a problem for the present manuscript but morphologic changes complicate the business in case of future application of the method. In the present version of the manuscript this is almost overlooked, apart from a quick mention at the end of the Conclusions. Few more lines on the issue would be an important addition.

RE: We agree in that morphological changes can really complicate the business in many practical cases. According to the suggestion, we changed the Conclusions to remark the possible limitations of assuming a fixed bed. Now the text reads:

> "A main limitation of the present methodological approach relies in the assumption of fixed bed in both the CFD analysis and the application of the entropic model. In natural rivers, bed scouring during sever flood events and the ensuing formation of local deposits, especially close to in-stream structures such as bridges, can alter the bathymetry and, in turn, the velocity distribution and the discharge estimates. In case of movable bed and absence of protection measures (e.g., riprap or bed sills), the uncertainty associated to the local bed mobility has to be evaluated with due care.
>
> Future research  on more complex scenarios that still need a comprehensive

assessment, and which could largely benefit from physics-based numerical modelling, will include the case of mobile beds and the analysis of stage-dependent variations of cross-sectional velocity distribution, particularly in case of compound cross-sections that are typical of natural rivers."

168: the events could be better described in terms of (i) flow over the sediment bar, (ii) pressurized flow below the arches, and (iii) bridge overflow. Later in the manuscript the reader will understand that only for the strongest event there was incipient pressurization of flow below the arches. It would be probably better to add details here.

RE: We added some details to the text, which now reads:

"In all the three flow conditions, water flowed in the main channel and over the sediment bars that are dry in the low flow condition of Figure 1b,d. During the most severe flood of 2012, water flowed on the floodplains adjacent to the main river and caused the incipient pressurization of flow below the bridge arches ."

173: is it possible to give some value of return period for these events?

RE: Yes. The return period associated to the three flood events are about 200 years for the 2012 event, 2 years for the 2019 event, and 1 year for the 2022 event. The information has been added to Table 1.

Tab 1: the discharge values come, I guess, from rating curves mentioned at line 133. Is it so?

RE: Yes. In the revised manuscript we have clarified this point.

Eq. (2) and (3): this is the first time M and phi(M) appear, but they are not given a name.

RE: Thank you for noting. Now the entropic function, $\phi(M)$, and the entropic parameter, $M$, are properly defined and referenced throughout the text.

Tab 2: please indicate how the values of phi(M) and M were obtained. From (3) and (2), respectively?

RE: Now it is clarified that the value reported in Table 2 of the previous version were obtained from the 3D-CFD velocity distributions. In the revised version, we added the values computed with the formula used to initialize the iterative procedure used in the entropy model. We also clarified how these values are computed, i.e., $\phi(M)$ from a formula for the first guess, then as the ratio of maximum to average velocity, and $M$ from $\phi(M)$ according to Eq. (2).

236: I would say this is true also for the 2019 event.

RE: Thank you for noting. The text was modified according to the suggestion.

294: I think that "accuracy" would be better than "precision" here. Same at line 366.

RE: We agree and substituted "precision" with "accuracy".

321: actually, in Tab 4 the errors for elliptic are larger than those for parabolic.

RE: We agree that the total discharge estimation appears more accurate for the PSD rather than ESD. However, the sentence was a comment on the cross-sectional distribution of the velocity for the undisturbed sections reported in the plots. This point is better specified in the revised version of the manuscript.

354: like the one mounted on this bridge (I mean the radar sensor).

RE: We added this specification to the revised text.

---

## Author Comment (AC2)

Reply to Comments by:

**REVIEWER 2**

**General Comments:**

The manuscript evaluates the efficacy of the two available approaches of estimating the velocity distribution of a given discharge passing at the Paglia River reach flow sections in the vicinity of the Adunata bridge subject to the conditions of with and without the impact of the presence of bridge piers on the velocity distributions of the flow estimated at these sections using the entropy theory. These velocity distributions are compared with that of the velocity distributions of the same discharge obtained at these respective flow sections based on the flow fields simulated by a Computational Fluid Dynamics (CFD) model, considered as the benchmark model. The CFD model is set up using different types of observed input data measured and collected based on the recorded water level, velocity measurements made using current meter, rating curves and free surface velocity data collected using the water level and velocity radar sensors mounted on the downstream of the Adunata bridge deck for the three considered past flow events of 2012, 2019 and 2022. Further, the velocity distributions estimated at the studied river flow sections based on the application of entropy theory were assessed using the ADCP measurements made at the section far away upstream of the Adunata bridge. To simulate the velocity distributions at the studied river sections using the entropy theory, the authors have adopted two approaches of using the surface velocities estimated by the CFD model simulations at the considered sections of the considered flow event, *viz*., the use of span-wise simulated surface velocity estimates at the considered river section, and use of only the simulated maximum surface velocity estimate at the same section. Based on the study, the authors arrive at the conclusion that the span-wise estimated surface velocity measurements are needed to effectively capture the sectional velocity distributions close to that simulated by the CFD model at the sections immediately downstream of the bridge where the flow fields are impacted by the disturbance generated by the presence of bridge piers; whereas, the flow sections at far away upstream and downstream of the bridge, where the flow fields of the passing discharge are not impacted by the bridge piers, the velocity distributions estimated at these section using the entropy theory based only on the use of maximum surface velocity information may be sufficient for closely reproducing the CFD model based velocity distributions at these sections.

The study is timely and a needed one to widen the knowledge on the field applicability of the entropy theory for discharge estimation. It can be inferred from the study that when the flow field in a river reach is not impacted by the presence of a structure constructed across a river, then the measurement of maximum surface velocity may be sufficient to estimate the discharge passing at that section using the entropy theory. However, when the river section flow characteristics are impacted by the presence of a structure constructed across the river, then span-wise surface velocity measurements may be required to simulate the actual velocity distribution that prevails at that section which is required for serving the purpose of studying scour around the bridge piers. Therefore, both these different approaches of estimating surface velocity measurements using velocity radar(s) have their relevance to serve their intended practical purposes. The manuscript deserves to be accepted for the stated reasons. However, the authors need to address many comments and incorporate corrections in the manuscript, as given in the following pages, before its publication in the final form.

RE: We thank the reviewer for appreciating the paper and for the constructive criticism and comments, which allow us to improve the paper.

**Specific Comments:**

(1) Since the manuscript describes the study carried out by the authors and then reported here, it would be appropriate to describe the study in the past tense rather than in the present tense, throughout the manuscript.

RE: Thank you for noting. Some parts of the text were already in the past tense. We reviewed the manuscript and moved to past other parts referring to the specific activities carried out in this study.

(2) Since the main emphasis of the study is related to the velocity distributions of a given discharge passing through many flow sections of the river reach in the vicinity of a bridge, the title of the manuscript may reflect on this aspect, specifically changed as "Estimating the velocity distribution and the discharge passing at different flow sections of a river reach in the vicinity of a bridge using the entropy theory: Insights from the flow fields generated by a computational Fluid Dynamics model."

RE: Thanks for the suggestion. We added "velocity distribution" in the title besides "flood discharge". We note that the title is already long, so it is desirable to add as little text as possible. We acknowledge that the cross-sections we used in the analysis are not located exactly at the bridge, but it is also the common practice that flow and discharge measurements are often carried out from bridge, but not exactly at the bridge section. The sections that are closest to the bridge, in our study, are 50 m upstream and downstream of the bridge, a distance which is about $0.45B$, with $B$ the width of the riverbed at the bridge section. These sections are indeed very close to the bridge, particularly considering that, while the water level sensors measure perpendicularly, the remote sensors for surface velocity (such as radar, Large Scale PIV, etc.) have their field of view located some tens of meter upstream or downstream of the bridge. Similarly, when a current meter is operated from a bridge, the drag force moves the current meter downstream at a distance that depends on the height of the bridge with respect to the free-surface elevation and on the drag force of the flowing water. Finally, note that the flow fields predicted by the CFD model show that the velocity distribution varies abruptly at the upstream edge of the bridge (flow contraction), and smoothly in the tens of meter upstream and downstream of the bridge. Accordingly, for the purpose of the study, we remain convinced that the particular choice of looking at cross-sections located 50 m upstream and downstream of the bridge can still be regarded as sections "at" the bridge. We are reluctant to use "in the vicinity" because of the length of the title (which was extended to include "velocity distribution"), and because it makes the title much less fluid than simply "at". We added some text at the beginning of Sect. 3 to explain the issue of cross-section spacing:

> "The sections just upstream and downstream of the bridge are located at a distance of about $0.45B$ from the bridge, with $B$ the width of the river at the bridge section. This is a short distance, particularly considering that the remote sensors for surface velocity (such as radar, Large Scale PIV, etc.) have their field of view located some tens of meter upstream or downstream of the bridge. The sections far downstream are considered to assess how far the flow field is affected by the presence of the bridge."

Finally, the fact that we also considered two cross-sections further downstream of the bridge is only meant to assess how the disturbances generated by the bridge vary in space. The core of the study is an attempt to answer the question: "how can we estimate the discharge with the entropic theory and measures carried out from river bridges?", which is well reflected by the modified title.

Thus, the title was modified to:
> "Estimating velocity distribution and flood discharge at river bridges using the entropy theory. Insights from Computational Fluid Dynamics flow fields"

(3) Line #38, Explain, what is the secondary current of the second kind?

RE: Secondary flows of the second kind are quite basic features of open-channel flows, which originate at the channel boundaries and streamwise corners because of turbulence heterogeneity. We added some words on this, just before the references in which the matter is clearly explained. Now the text reads:

> "…the presence of banks and of discontinuities of bed elevation in the spanwise directions can generate secondary currents of the second kind because of turbulence heterogeneity (Nikora and Roy, 2011; Proust and Nikora, 2020)."

(4) Line #65, use of some field data! Which field data?

RE: Sorry. We agree that the text was not clear (and annoying too). Indeed, we rewrote and improved large part of the Introduction. In the revised text, now we first introduce different techniques used measure the surface velocity, and then of methods the use these data to estimate the flow discharge.

(5) Line #76, What do you mean by "weak gauging sites?" or is it wake affected gauging sites?

RE: We intended a gauging site with insufficient data to obtain reliable estimates of the flow discharge. This part of the text has been deleted to improve the readability of the Introduction.

(6) Line #89, two European rivers! Specify these two rivers.

RE: In the revised text, this sentence has been removed.

(7) Lines #106-107, severe flooding and high sediment transport! What is the impact of the high sediment rate transport on the flow velocity and its distribution under different magnitudes of sediment laden flood discharge?

RE: The issue of sediment transport and of mobile bed has been also raised by the first reviewer. While the problem of bed mobility is somewhat of second order in view of estimating the flood discharge using non-contact sensors, we agree in that solid transport and morphological changes can really complicate the business in many practical cases. In the specific case considered in the present study, the bed is stabilized by a bed sill about 300 m downstream of the bridge. Historical aerial views show minor changes in the point bar forming downstream of the bridge on the left, reasonably because the bridge structure acts to stabilize the riverbed from (at least) a planimetric point of view. According to the reviewer suggestion, we changed the Conclusions to better remark the possible limitations of assuming a fixed bed. Now the text reads:

> "A main limitation of the present methodological approach relies in the assumption of fixed bed in both the CFD analysis and the application of the entropic model. In natural rivers, bed scouring during sever flood events and the ensuing formation of local deposits, especially close to in-stream structures such as bridges, can alter the bathymetry and, in turn, the velocity distribution and the discharge estimates. In case of movable bed and absence of protection measures (e.g., riprap or bed sills), the uncertainty associated to the local bed mobility has to be evaluated with due care.
>
> Future research  on more complex scenarios that still need a comprehensive

assessment, and which could largely benefit from physics-based numerical modelling, will include the case of mobile beds,  and the analysis of stage-dependent variations of cross-sectional velocity distribution, particularly in case of compound cross-sections that are typical of natural rivers."

(8) Line #117, "mayo axis", is it "major axis"?

RE: Yes, of course. We fixed the typo. Thank you.

(9) Line #167, it is stated that three different steady flow conditions are simulated using the 3D-CFD model which correspond to the peak flow conditions of flood events occurred in 2012, 2019 and 2022. But in Line #173, it is stated that surface velocity data are not available for the 2012 flood event. So how the 2012 flow event's velocity distribution was simulated using the CFD model for the peak discharge of the 2012 flow event?

RE: We agree that, in the previous version, the use of data (either measured or derived from the 3D-CFD numerical model) was not sufficiently clear, and apologize for that. As a preliminary step, a 2D depth averaged model was setup and calibrated with the rating curve available for the bridge section (Appendix A), in order to obtain reliable water level elevations downstream of the bridge. A 3D-CFD model was then setup and forced with the flow discharge at the inlet and the water stage (derived from the 2D model) at its outlet. The CFD model, which does not require any calibration, was validated using the surface velocity data that were available for the 2019 and 2022 event (Appendix A), not for the 2012 flood event (the instrument was not mounted at that time yet). The 3D-CFD model has been used to derive physics-based flow fields in the vicinity of the bridge. Of the 3D-CFD flow fields, extracted at the four considered cross-sections, the surface velocity distributions were used as input for the entropy model, and the cross-sectional velocity distributions were used to benchmark the results of the entropy model.
The text was changed and improved so that we are confident that in the revised version of the manuscript these aspects are sufficiently clear.

(10) Line #185, define phi(M)!

RE: Thank you for noting. Now all the parameters and functions are properly introduced and defined. Please see also our answer to major point #2 by Reviewer 1.

(11) Line #208, Phi(M) is defined as entropy parameter, and in Line# 74, the parameter M is also defined as entropy parameter. So, a consistent definition of these parameters needs to be given.

RE: Thank you for noting. As reported in the comment above, now all the parameters and functions are properly introduced and defined. $\phi(M)$ is the entropy function and $M$ is the entropy parameter.

(12) Table-2 contents, the second line inside the Table the estimate of M is given as -1.03, what is the physical meaning of a negative M?

RE: The $M$ value is the main parameter the entropic distribution depends on. According to Eq. (2) in the paper, $M$ has a one-to-one relationship with the entropic function, $\phi(M)$, which, being the ratio of the average to the maximum velocity in the cross-section, is always positive. The physical meaning of the entropy function, $\phi(M)$, is clear, as the values it assumes are larger (i.e., close to one) when the velocity distribution is nearly uniform. The magnitude of $M$ can be negative depending on the analytical nature of Eq (2). Overall, it usually happens when the distribution of the cross-sectional flow is far from being uniform, i.e., when the maximum velocity within the

cross-section is much larger than its average value. In the present study, this occurs just downstream of the bridge, where the abrupt geometrical variations produce strong acceleration in the flow field.

---

## Author Response (AR2)

Review of "Estimating velocity distribution and flood discharges at river bridges using the entropy theory. Insights from Computational Fluid Dynamics flow fields" by Bahmanpouri, F. et al.

**REVIEWER 1**

Thanks to the authors of this manuscript for accounting for the comments I provided. I think that improving the explanation of how the method is applied is particularly beneficial for the manuscript. I have just two remaining comments (the second of which is a bit more challenging than the first) and several suggestions for change of wording at specific places. Please note, line numbers refer to the track-change version of the submission.

RE: We thank the reviewer for appreciating the revision work.

58: this line sounds like horseshoe and wake vortices are created by secondary currents; rather, they form due to the flow-bridge interaction. I suggest rephrasing.

RE: Thank you for pointing out. We reworded the sentence to
> " Systems of vortices with horizontal (horseshoe vortex) or vertical axes (wake vortex)  modify the velocity distribution"

301: these lines introduce a concept that, if I remember correctly, was not mentioned in the previous version. Attributing to an increased Manning coefficient the energy dissipation that is instead due to Reynolds stresses may work, but then the "turbulence-enhanced" Manning coefficient is a new parameter. So, two questions: (1) was the increased value of 0.055 the result of a calibration? (2) What should one do to apply (4) in a wake region, are there guidelines?

RE: We agree with the reviewer in that the "turbulence enhanced" Manning coefficient is indeed a new parameter. We observe that, while it is quite common to use a higher Manning coefficient to account for additional head losses due to the presence of in-stream structures, the purpose of using here a larger value is not determinant for the application of the (iterative) procedure. It was only meant to show that the entropic function may assume similar values to those computed from the 3D-CFD flow field by accounting for additional head losses at the cross-section just downstream of the bridge. This is expressed in the comment added to respond to the next point.

305: after presenting tab 2, a comment is probably needed about the similarity of the M and phi(M) values obtained with the two approaches. The intent declared at line 311 does not result in a statement (using, for example, the two panels of fig. 3).

RE: Thank you for noting. We added a comment on the similarity of the M and phi(M) values obtained with the two approaches. In the same comment, we also frame the additional calibration effort needed to obtain the Manning values in the case of disturbed flows (see the previous comment):
> "The first-guess estimates of $\phi(M)$ in Figure 3b, although having a marginal role in the entropy-based computations, show a similar trend to the 3D-CFD estimates (Figure 3b),

provided that increased Manning parameter is used at the section just downstream of the bridge. The need to calibrate such an increased Manning parameter complicates efforts in case of disturbed flows."

Minor Comments

19: downstream of

23: suggests

33: applicable

38: no new line here

47 to making

124: is used

261: denotes

500: severe

504: no new line here

RE: All the above requests for changes have been applied. Thank you.

**REVIEWER 2**

I found this paper interesting. The topic of Information Entropy is very actual and I think that the entropy-based method can be applicable to many issues of risk assessment and environmental research.

RE: We thank the reviewer for appreciating the work.

The main criticism I found in the paper is the calculation of the entropy-parameter M. The authors state that that Um in the equation (2) is the average flow velocity within entire cross-section.

Such statement is correct only when 1D distribution is considered in the form of wide channel, according to Chiu's initial work (1987, 1988). In general, Um in equation (2) represents the expected value of velocity. Marini and Fontana (2020) have clarified this aspect.

This issue affects the model's outcome because using the mean velocity value instead of the expected value of velocity in equation (2) leads to an incorrect calculation of M. Consequently, the resulting velocity distribution will have an average flow velocity different from the desired one.

For example, consider the calculations performed for the "2019 +50" configuration. Referring to the data in Table 2 the average flow velocity is 1.93 m/s. Now setting in eq. (2) the expected value of velocity $U_m$=1.93 m/s results in F(M)=0.515 and therefore M=0.18. With M=0.18, applying equation (1) coupled with equation (5) will yield an average flow velocity 1.93 m/s. If the channel were wide, as demonstrated by Marini and Fontana (2020), the mean sectional velocity would be greater than 1.93m/s. This inconsistency should be apparent in the authors' results; in fact, I would have expected an average flow velocity reported in Table 3 different from 1.93. However, the authors report in Table 3 that the

result of the average flow velocity calculated by integrating the velocity distribution values is 1.90, practically the same as the expected one. Therefore, the inconsistency does not appear.

The reason this inconsistency does not appear, in my opinion, is that the authors use an iterative procedure to find the velocity distribution based on controlling the error in calculating F(M) between successive iterations. This iterative procedure is necessary because the authors impose that the M value for different verticals of the same river cross-section must be the same. However, I believe that the main reason this procedure must be used is to obtain a mean flow velocity value equal to the desired one.

I would like to clarify that I am not saying the authors' procedure is incorrect (Moramarco is an undisputed authority in the entropy-hydrology field), but I believe the authors should emphasize:
- The difference between the expected value of velocity (which appears in equation (2) - the Chiu original one) and the average flow velocity, which is considered given in the problem.
- Due to the discrepancy between the expected value of velocity and the average flow velocity, an iterative process is necessary. This process seeks the value of Um (expected value) which, when inserted into equation (2), provides an M value from which a velocity distribution can be obtained that has the desired average flow velocity.
At this point, a column for the expected velocity values Um for each configuration, obtained as a result of the iterative procedure, should be added to the results tables (Table 2). This would make it clear that Um is not a known a priori value (the known a priori value is the average flow velocity) but a value calculated through a procedure.
Gustavo Marini

RE: We thank Prof. Marini for his interesting point. In accordance with his suggestion, we added a comment in the text recalling the difference between the expected value of velocity and the average velocity and quoting Marini & Fontana (2020). In our case, however, considering the aspect ratio of the river channel greater than 5 (as mentioned in Table 2), these two values are quite similar.
As described at the end of Sect. 2, in applying the Entropy model we used as input value only the cross-section geometry and the surface velocity (either river-wide or a single value), then we computed the expected velocity and, hence, the discharge using the bathymetry-based flow area. Importantly, it has to be considered that we forced the Entropy model with a variable spanwise distribution of both the bathymetry and the surface velocity. This is true in particular for the +50 cross-section, just downstream of the bridge, where the velocity field is strongly perturbed by the bridge piers, thus markedly irregular in the spanwise direction. As we used a single $M$ value for all the verticals in a single cross section, the iterative procedure is used to make the spanwise entropic distribution coherent with the definition of $\phi(M) = U_m/U_{MAX}$ given in Eq. (2). As regards the hypothesis to fit $M$ through an iterative approach, which minimizes the error between expected $U_m$ and the average velocity, this is beyond the paper's purpose but will be the object of future work definitely.
Finally, as per the request of adding a column with the velocity values Um for each configuration, obtained as a result of the iterative procedure, we note that these values of Um are already reported in Table 3.
In the revised version of the manuscript, we added a sentence just after Eq. (2) to warn about the possible difference between the average and the expected value of the velocity. The added sentence reads:

"It is worth mentioning that, Um represents the expected value of velocity that can be different from the observed mean velocity (Marini and Fontana, 2020). These two values are quite similar in the case of wide rivers (aspect ratio larger than 5). In the present research, considering the large aspect ratio for all cross-sections (Table 2), this hypothesis is valid."

Added reference

Marini, G. and Fontana, N.: Mean Velocity and Entropy in Wide Channel Flows, Journal of Hydrologic Engineering, 25, 06019009, https://doi.org/10.1061/(ASCE)HE.1943-5584.0001870, 2020.